# Deletion of *Stk11* and *Fos* in mouse BLA projection neurons alters intrinsic excitability and impairs formation of long-term aversive memory

David Levitan[1†]*, Chenghao Liu[1†], Tracy Yang[1], Yasuyuki Shima[1], Jian-You Lin[2,3], Joseph Wachutka[2], Yasmin Marrero[2], Ramin Ali Marandi Ghoddousi[1], Eduardo da Veiga Beltrame[2], Troy A Richter[1], Donald B Katz[2,3], Sacha B Nelson[1,3]*

[1]Departments of Biology, Brandeis University, Waltham, United States; [2]Departments of Psychology, Brandeis University, Waltham, United States; [3]Volen Center for Complex Systems, Brandeis University, Waltham, United States

**Abstract** Conditioned taste aversion (CTA) is a form of one-trial learning dependent on basolateral amygdala projection neurons (BLApn). Its underlying cellular and molecular mechanisms remain poorly understood. RNAseq from BLApn identified changes in multiple candidate learning-related transcripts including the expected immediate early gene *Fos* and *Stk11*, a master kinase of the AMP-related kinase pathway with important roles in growth, metabolism and development, but not previously implicated in learning. Deletion of *Stk11* in BLApn blocked memory prior to training, but not following it and increased neuronal excitability. Conversely, BLApn had reduced excitability following CTA. BLApn knockout of a second learning-related gene, *Fos*, also increased excitability and impaired learning. Independently increasing BLApn excitability chemogenetically during CTA also impaired memory. STK11 and C-FOS activation were independent of one another. These data suggest key roles for *Stk11* and *Fos* in CTA long-term memory formation, dependent at least partly through convergent action on BLApn intrinsic excitability.

**\*For correspondence:**
levitand@brandeis.edu (DL);
nelson@brandeis.edu (SBN)

[†]These authors contributed equally to this work

## Introduction

Conditioned taste aversion (CTA) is a form of long-lasting aversive memory induced by a single pairing of exposure to an initially palatable taste with gastric malaise (*Bures et al., 1998*). Although multiple brain regions, including the brainstem, amygdala and the cortex, participate in various aspects of taste behavior (reviewed in *Carleton et al., 2010*), prior work suggests that the basolateral amygdala (BLA) plays a critical role in CTA memory. Disrupting neuronal activity within the BLA blocks the formation and retrieval of CTA memory (*Yasoshima et al., 2006*; *Ferreira et al., 2005*; *Garcia-Delatorre et al., 2014*; *Molero-Chamizo and Rivera-Urbina, 2017*). This may reflect the fact that BLA projection neurons (BLApn) provide the principal output pathway from the amygdala to fore-brain structures including the gustatory cortex and the central amygdala (*Duvarci and Pare, 2014*) enabling it to distribute taste valance information to these regions (*Piette et al., 2012*; *Samuelsen et al., 2012*). Consistent with this view, BLA neurons change their activity and their functional connectivity with their down-stream targets during CTA learning (*Grossman et al., 2008*). However, whether the BLA is a site of cellular and molecular plasticity during CTA learning, as opposed to merely gating plasticity in other structures, is not known.

Stages of memory formation are typically distinguished on the basis of duration and molecular mechanism. Short-term memory, lasting minutes to hours, requires only post-translational

modification of preexisting proteins, whereas long-term memory, lasting days or longer, requires gene transcription and RNA translation, typically occurring in the hours following memory acquisition (*Matthies, 1989*; *Alberini, 2009*; *Gal-Ben-Ari et al., 2012*; *Kandel, 2001*). Production of new proteins is required to produce lasting changes in the efficacy of synaptic connections and in the intrinsic excitability of neurons, which are thought to be the cellular correlates of memory (*Zhang and Linden, 2003*; *Mozzachiodi and Byrne, 2010*). The cellular correlates of CTA learning are less completely understood than those of some other forms of learning, but the involvement of both synaptic plasticity (*Li et al., 2016*) and intrinsic plasticity (*Yasoshima and Yamamoto, 1998*) have been demonstrated. CTA is known to require protein synthesis in the BLA (*Josselyn et al., 2004*) and to increase the expression of the activity-dependent transcription factor C-FOS (*Uematsu et al., 2015*). In other behavioral paradigms, neurons increasing C-FOS protein undergo changes in synaptic strength and intrinsic excitability (*Yassin et al., 2010*; *Ryan et al., 2015*; *Pignatelli et al., 2019*) and are thought to be essential parts of the neuronal network underlying long-term memory (*Tonegawa et al., 2015*). However, the role of neurons expressing C-FOS in the BLA during CTA is unclear. Also unknown is whether CTA learning requires new transcription, and if so, the identities of the required transcripts and cellular processes they promote are not known.

In this study, we found that new transcription in the BLA is required for CTA learning. Using RNA-seq from sorted neurons, we found that CTA changed the expression of many genes. In addition to the expected activation of *Fos*, expression of the kinase *Stk11*, also known as *Lkb1*, is altered following learning in BLA projection neurons (BLApn), but not in excitatory or inhibitory neurons within the GC. STK11 is known to act as a master regulator of growth, metabolism, survival and polarity by phosphorylating 13 down-stream members of the AMP-related kinase family (*Lizcano et al., 2004*; *Shackelford and Shaw, 2009*). Recent work also suggests roles for STK11 in the nervous system, where it controls axonal specification and dynamics during development (*Barnes et al., 2007*; *Shelly et al., 2007*) and synaptic remodeling during old age (*Samuel et al., 2014*). STK11 can also regulate synaptic transmission in forebrain neurons (*Kwon et al., 2016*), but it has not previously been studied in the context of learning and memory or in the regulation of intrinsic neuronal excitability.

We find that *Stk11* is required for CTA since conditional knockout from BLApn prior to training impairs learning. However, the same deletion performed 2 days after training—that is, at a time when long-term memories have already been formed and stabilized—has no effect on subsequent memory retrieval. Deletion of *Stk11* also increased the excitability of BLApn, but did not alter the ability of the BLA-GC circuit to become transcriptionally activated by training or to encode the palatability of gustatory stimuli. CTA training is associated with an opposing decrease in intrinsic excitability in BLA projection neurons expressing C-FOS following learning. BLApn knockout of *Fos* also increased excitability and impaired learning, and expression of C-FOS and STK11 did not depend on each other. Thus, two independent genetic manipulations that affect learning also increase excitability. Using chemogenetics to independently increase BLApn excitability during CTA training without knocking out either gene also impaired CTA memory. Together, these data identify key roles for *Stk11* and *Fos* in CTA long-term memory formation and suggest they may act by convergently modulating the intrinsic excitability of BLApn.

## Results

### CTA long-term memory requires BLA transcription

In order to determine whether CTA requires new RNA transcription within the BLA, we inhibited transcription by injecting Actinomycin-D (1 µl, 50 ng, bilaterally, *Figure 1*), a widely used RNA polymerase two inhibitor (*Alberini, 2009*), into the BLA 20 min prior to CTA training, and tested memory 48 hr later. As a control, a separate group of mice received vehicle injection (1 µl of PBS, bilaterally). CTA training consisted of 30 min of access to 0.5% saccharin followed by an intraperitoneal injection of 0.15M LiCl, 2% body weight; (*Figure 1—figure supplement 1*). A two-way mixed ANOVA comparing vehicle and actinomycin-D injected mice before and after training revealed a significant effect of treatment and a significant interaction between treatment and training (*Figure 1B*). Post-hoc analysis revealed significant reduction in the consumption of saccharin (CTA vs. Test) only for the vehicle group, indicating impairment of learning in the actinomycin-D treated mice. As a convergent

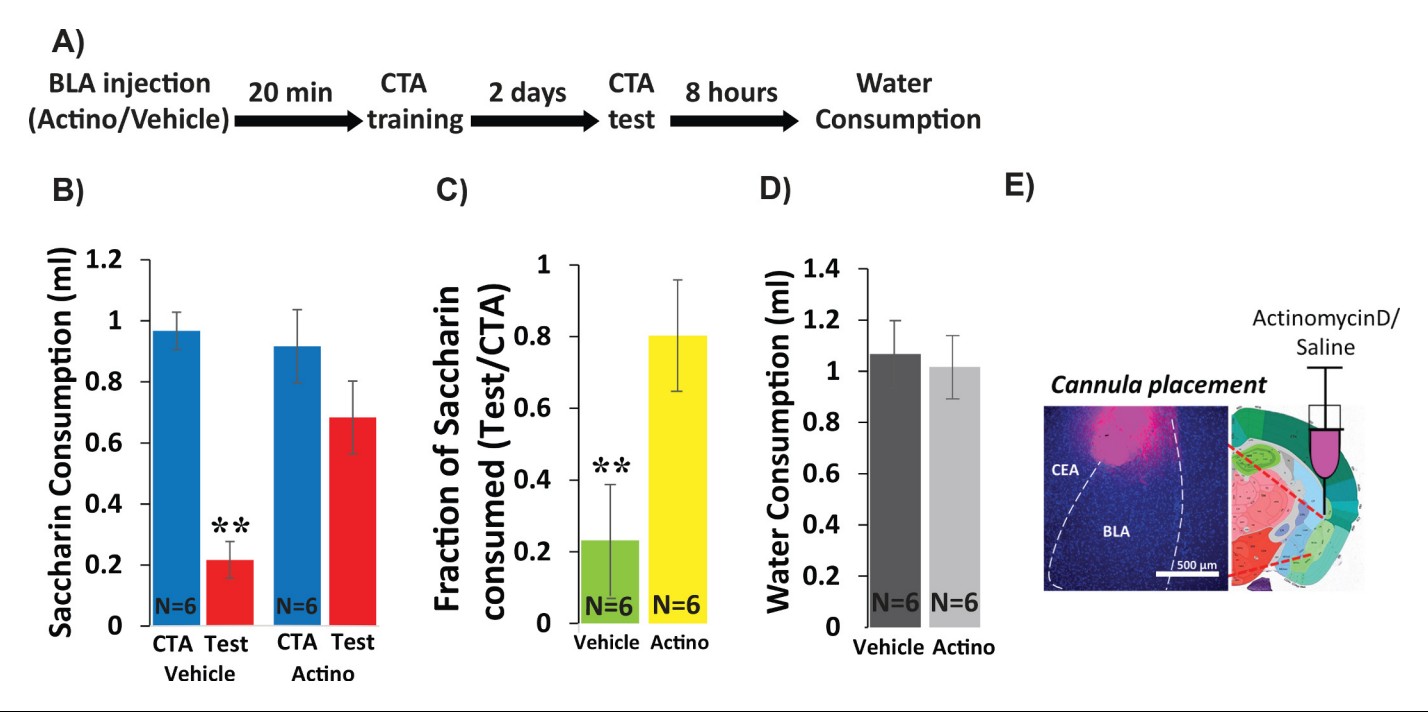

**Figure 1.** Inhibiting BLA transcription impairs CTA Learning. (A) Protocol for injection (1 μl per hemisphere) of Actinomycin-D (50 ng) or vehicle (PBS with 0.02% DMSO). (B) Actinomycin D injection prior to CTA training impairs learning, expressed as a reduction in saccharin consumption between CTA and test sessions. Two-way mixed ANOVA revealed a significant effect of treatment and a significant interaction between training and treatment (treatment: $F_{(1,10)} = 9.99$, p=0.010; training: $F_{(1,10)} = 4.62$, p=0.057; interaction: $F_{(1,10)} = 5.237$, p=0.045). Post hoc analysis (Bonferroni corrected) revealed significant reductions (p=0.001) of saccharin consumption (training vs. test) for vehicle treated, but not for actinomycin-D treated mice (p=0.94), and significant differences between the treatment groups during the test (p=$7.1\times10^{-3}$) but not during the training session (p=0.56). (C) Fraction of saccharin consumed (Test/CTA) was significantly higher (t(10)=-3.4; p=0.007) following Actinomycin D treatment than vehicle, consistent with weaker memory. (D) Treatments did not differ in water consumption measured 8 hr later (t(10)=0.28; p=0.79) suggesting this does not account for differences in consumption during the test. **p<0.01. (E) Guide cannula was coated with fluorescent dye to assess placement (Left) relative to desired location in anterior BLA (Right; bregma −1.4 mm; Allen brain atlas). Note that the injection cannula extended 0.5 mm further into the BLA. See also *Figure 1—source datas 1* and *2* and *Figure 1—figure supplement 1*.

The online version of this article includes the following source data and figure supplement(s) for figure 1:

**Source data 1.** Saccharin consumption (ml) during CTA training and test.
**Source data 2.** Fraction of saccharin consumed (Test/Training).
**Figure supplement 1.** Testing CTA in mice.

measure, we also assessed the strength of CTA memory by calculating the relative consumption of saccharin during the test day to that consumed on the training day (*Neseliler et al., 2011*). The differences between the groups (23% in the vehicle group vs. 80% for the actinomycin D group) were significant. Meanwhile, actinomycin-treated mice were neither impaired in their ability to detect the palatability of saccharin, nor in their drinking behavior—consumption of saccharin during CTA training was similar for the two groups, as was consumption of water 8 hr after the test (*Figure 1D*), suggesting that these nonspecific effects cannot account for the memory impairment. Thus, BLA transcription is essential for CTA memory formation. These results extend prior work showing the importance of BLA protein synthesis for CTA memory (*Josselyn et al., 2004*) and together show that training induces both transcription and translation important for CTA in the BLA.

## Transcripts regulated in BLA projection neurons following CTA

In order to identify specific transcripts that might be necessary for CTA learning within the BLA, we used cell-type-specific RNA-seq to profile transcriptional changes in sorted BLA projection neurons (BLApn). We manually isolated fluorescently labeled BLApn neurons from YFP-H mice (*Feng et al., 2000*) in which YFP is expressed under the Thy1 promoter in a large population of excitatory

projection neurons located in the anterior part of the nucleus (*Sugino et al., 2006*; *Jasnow et al., 2013*; *McCullough et al., 2016*). RNA sequencing was performed separately on YFP$^+$ BLApn harvested 4 hr following training from mice undergoing CTA, and from taste-only controls (n = 4/group) (*Figure 2* and *Table 1*). A single control condition consisting of the taste stimulus without the unconditioned stimulus was used since our principal aim was identify target genes that could then be tested for causal roles in learning in subsequent behavioral experiments and since a prior study *Rappaport et al., 2015* used a microarray screen following CTA to show that LiCl treatment (compared to taste control treatment) induced more than half of the same genes induced by CTA treatment (compared to taste control treatment).

Sequencing results (*Figure 2A*) also confirmed the purity and molecular identity of YFP-H neurons in the BLA, as transcripts known to be expressed in BLApn and other forebrain excitatory projection neurons were enriched and transcripts known to be expressed in inhibitory interneurons, glia cells and other neurons in the vicinity of the BLA (lateral amygdala or central amygdala) were virtually absent. Moreover, this observed pattern of expression was comparable to that reported in other studies profiling the same population (*Sugino et al., 2006*; *McCullough et al., 2016*).

CTA training changed expression of many genes. *Table 1* lists the 20 transcripts showing differential expression between CTA and taste control group based on robust expression criteria: $2 \leq$ fold change$\leq 0.5$, p<0.01 (unpaired t-test), transcripts-per-million (TPM) $\geq 30$. Included among these transcripts is the activity-dependent transcription factor, *Fos* which has previously been shown to be upregulated in BLApn following CTA and other learning paradigms (*Zhang et al., 2002*; *Yasoshima et al., 2006*; *Mayford and Reijmers, 2015*; *Uematsu et al., 2015*). *Fos* is an activity-dependent transcription factor that controls the transcription of downstream targets, which are capable of changing neuronal physiology (*Malik et al., 2014*). Among these, YFP$^+$ BLApn, express Nptx2, a member of the pentraxin family involved in AMPA receptor clustering (*Chang et al., 2010*). Our sequencing results revealed a two-fold increase of Nptx2 levels following CTA (fold-change = 2.1; p=0.029; N = 4/group, unpaired t –test), which was further validated by qPCR in separate experiments (Fold-change = 2.4; p=0.028; N = 4/group; *Figure 2—figure supplement 3*).

Our screen for differentially expressed genes also identified *Stk11* (also known as *Lkb1*; *Figure 2B*) a kinase well studied in the contexts of cancer, cell growth and development, but not previously studied in the contexts of learning and neuronal plasticity (*Bardeesy et al., 2002*; *Alessi et al., 2006*; *Barnes et al., 2007*; *Gurumurthy et al., 2010*; *Courchet et al., 2013*). Changes in the expression of both *Fos* and *Stk11* were validated by qPCR in separate experiments, which revealed a 4.6-fold increase in *Fos* mRNA (t(6)=2.5; p=0.045) and a 1.9-fold decrease in *Stk11* (t(6)=-2.5; p=0.046) (*Figure 2C*). To further examine the significance of these transcriptional changes, we also analyzed the levels of FOS and STK11 protein in the BLA. The fraction of C-FOS-expressing YFP-H neurons in the BLA 4 hr following CTA was significantly increased relative to lithium chloride-only and taste-only control groups (*Figure 2—figure supplement 1*). We measured STK11 4 hr following CTA or presentation of taste alone, by performing immunoblotting of proteins isolated from the anterior BLA. Surprisingly, we found a 1.8-fold increase (*Figure 2—figure supplement 2*) indicating that protein and mRNA expression are regulated in opposite directions at this time point.

STK11 is at the apex of the AMP-related kinase pathway and mediates its effects by phosphorylating one or more of 13 different downstream kinases, all of which share some homology with AMP-kinase (*Lizcano et al., 2004*; *Shackelford and Shaw, 2009*). Among these, YFP$^+$ BLApn had moderate expression of *Brsk2* (TPM = 39.9) and *Mark2/3* (TPM = 59.3 and 73.7 respectively; *Figure 2—figure supplement 3*), which have known roles in establishing cell polarity during neuronal development (*Barnes et al., 2007*; *Shackelford and Shaw, 2009*). Expression of AMP kinase itself (*Prkaa1/2*), an important metabolic regulator in many cell types (*Shackelford and Shaw, 2009*) was lower (TPM = 16.4 and 22.3, respectively). Comparing kinase expression between CTA and control groups revealed nearly two-fold less expression of *Mark2* following CTA (fold-change = 0.41; p=0.04, unpaired t-test; data not shown), but this was not significant after Bonferroni correction across the compared kinases.

We also analyzed the impact of CTA on the transcriptional profile of layer five pyramidal neurons labeled in strain YFP-H and Pvalb-expressing inhibitory interneurons in the GC, a brain region in which transcription is also known to be important for CTA learning (*Inberg et al., 2016*). RNA sequencing was performed on YFP$^+$ and Pvalb+ neurons in the GC harvested from mice 4 hr following CTA training, and from taste-only controls (n = 3/4/group). *Tables 2* and *3* list the transcripts

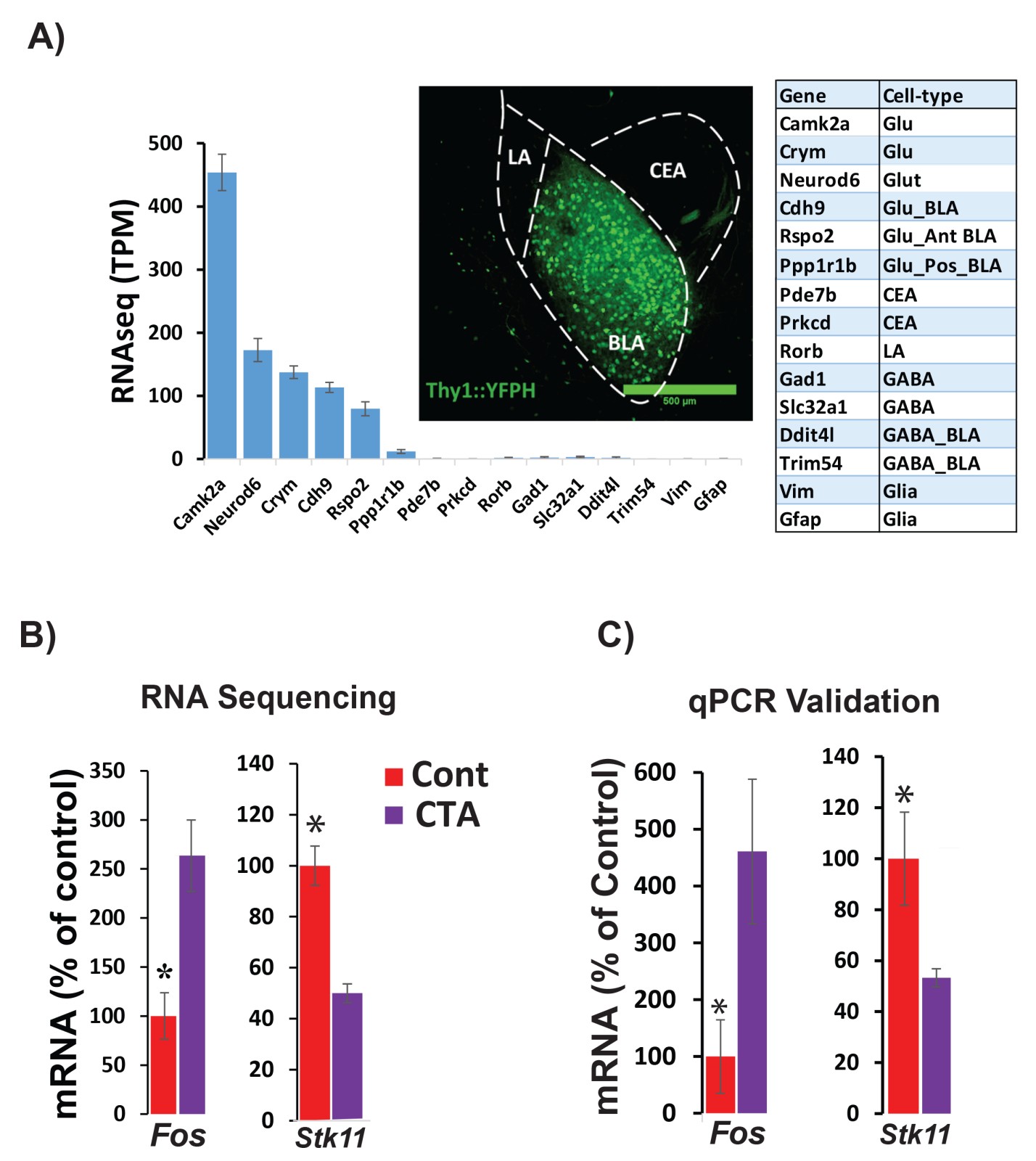

**Figure 2.** RNA sequencing from BLApn 4 hr following CTA training. (**A**) BLApn were isolated from YFP-H mice following CTA or taste control (N = 4/group). Neurons (150-200) were manually sorted form coronal slices (LA-lateral amygdala; CEA-central amygdala). Abundant transcripts (histogram, averaged across both groups) are enriched for those expected in the population and depleted for those expressed in other nearby populations (*Table 1*) including GABAergic interneurons, glia, or neurons in LA or CEA; *Sugino et al., 2006*; *Kim et al., 2016*; Allen brain atlas). Glu- GABA-,

*Figure 2 continued on next page*

*Figure 2 continued*

glutamatergic, GABAergic neurons; AntBLA, PostBLA- Anterior and posterior portions of the BLA. TPM- transcript per million. **(B)** Among genes meeting robust criteria for differential expression (see *Table 1*) *Stk11* and *Fos* were selected for further analysis, including qPCR confirmation **(C)** in separate experiments (N = 4/group; *p<0.05). See also *Figure 2—source data 1* and *Figure 2—figure supplements 1–4* and *Tables 1–3*. The online version of this article includes the following source data and figure supplement(s) for figure 2:

**Source data 1.** qPCR validation for Fos and Stk11 mRNA expression in YFPH+ neurons.
**Figure supplement 1.** CTA increases C-FOS protein expression in BLApn including those in strain YFP-H.
**Figure supplement 1—source data 1.** FOS protein counts in BLA 4 hr following CTA training and LiCl control.
**Figure supplement 1—source data 2.** FOS protein counts in BLA YFPH+ neurons 4 hr following CTA training and LiCl control.
**Figure supplement 2.** STK11 protein expression following CTA training.
**Figure supplement 2—source data 1.** STK11 protein levels in the BLA 4 hr following CTA training or taste control.
**Figure supplement 3.** BLApn express known downstream targets of STK11 and C-FOS.
**Figure supplement 4.** RNA sequencing from GC.

showing the most differentially expressed genes using the same criteria used in the BLA: $2 \leq$ fold change$\leq 0.5$, p<0.01 (unpaired t-test), transcripts-per-million (TPM) $\geq 30$. While Pvalb+ neurons

**Table 1.** Transcripts in YFP$^+$ BLApn with significantly altered expression 4 hr following CTA Criteria: $2 \leq$ Fold Change$\leq 0.5$, p<0.01, TPM >30 (TPM = transcript per million).

| Symbol | Fold- Change | P-Value | Gene name |
|---|---|---|---|
| **Upregulated in CTA vs taste control** | | | |
| Nptx1 | 2.02 | 1.57E-4 | Neuronal pentraxin 1 |
| Ric8 | 2.06 | 0.0017 | RIC8 guanine nucleotide exchange factor A |
| Mmab | 3.66 | 0.0019 | Methylmalonic aciduria (cobalamin deficiency) cblB type homolog |
| 1110008F13Rik | 2.08 | 0.0028 | RAB5 interacting factor |
| Kbtbd4 | 2.09 | 0.0044 | Kelch repeat and BTB (POZ) domain containing 4(Kbtbd4) |
| Nudt21 | 2.14 | 0.0077 | Nudix (nucleoside diphosphate linked moiety X)-type motif 21 |
| Fos | 2.64 | 0.0094 | FBJ osteosarcoma oncogene |
| Magoh | 2.72 | 0.0095 | Mago homolog, exon junction complex core component |
| **Downregulated in CTA vs taste control** | | | |
| Surf2 | 0.40 | 5.42E-4 | Surfeit gene 2(Surf2) |
| Tmem136 | 0.39 | 8.69E-4 | Transmembrane protein 136 |
| Stk11 | 0.50 | 0.0011 | Serine/threonine kinase 11(Stk11) |
| Kank3 | 0.28 | 0.0022 | KN motif and ankyrin repeat domains 3 |
| Lrrn1 | 0.49 | 0.0024 | Leucine-rich repeat protein 1, neuronal |
| Trpc1 | 0.48 | 0.0032 | Transient receptor potential cation channel, subfamily C, memb. 1 |
| Prpf6 | 0.49 | 0.0045 | Pre-mRNA splicing factor 6 |
| Tctex1d2 | 0.49 | 0.0052 | Tctex1 domain containing 2 |
| Gpr108 | 0.37 | 0.0056 | G-protein-coupled receptor 108 |
| Vkorc1 | 0.39 | 0.0069 | Vitamin K epoxide reductase complex, subunit 1 |
| D10Wsu102e | 0.50 | 0.0082 | DNA segment, Chr 10, Wayne State University 102, expressed |
| Tmem107 | 0.28 | 0.0089 | Transmembrane protein 107 |

**Table 2.** Transcripts in YFP⁺ L5 pyramidal neurons in the GC with significantly altered expression 4 hr following CTA
Criteria: 2 ≤ Fold Change≤0.5, p<0.01, TPM >30.

| Gene symbol | Fold-change | p-Value | Gene name |
|---|---|---|---|
| | | | Transcript Down-regulated CTA vs taste control |
| Exosc1 | 0.43 | 0.008 | Exosome component 1 |

showed a robust transcriptional response, evident by 19 genes reaching the criteria, only one gene reached the same criteria in YFP+, suggesting a weaker CTA-driven transcriptional response in these neurons. Importantly, the expression of *Fos* and *Stk11* in both YFP+ and Pvalb+ neurons, did not differ between CTA and control groups (*Figure 2—figure supplement 4* and *Tables 2* and *3*). This suggests that the differential expression of *Fos* and *Stk11* during CTA learning may be specific to a subset of cell-types within the circuit.

**Table 3.** Transcripts in Pvalb⁺ interneurons in the GC with significant altered expression 4 hr following CTA
Criteria: 2 ≤ Fold Change≤0.5, p<0.01, TPM >30.

| Symbol | Fold-change | p-Value | Gene name |
|---|---|---|---|
| **Upregulated in CTA vs taste control** | | | |
| Uprt | 3.38 | 0.001 | Uracil phosphoribosyltransferase |
| Snca | 2.66 | 0.001 | Synuclein, alpha |
| 1810043H04Rik | 2.57 | 0.004 | NADH:ubiquinone oxidoreductase complex Assemb.Fact. 8 |
| Dedd | 2.47 | 0.001 | Death effector domain-containing |
| Fam149b | 2.26 | 0.007 | Family with sequence similarity 149, member B |
| Jazf1 | 2.12 | 0.008 | JAZF zinc finger 1 |
| Nup54 | 2.09 | 0.007 | Nucleoporin 54 |
| **Downregulated in CTA vs taste control** | | | |
| Dear1 | 0.006 | 0.0006 | Dual endothelin 1/angiotensin II receptor 1 |
| Nt5c3b | 0.20 | 0.002 | 5'-nucleotidase, cytosolic IIIB |
| Lrrc16b | 0.26 | 0.010 | Capping protein regulator and myosin 1 linker 3 |
| Enc1 | 0.31 | 0.005 | Ectodermal-neural cortex 1 |
| Asap2 | 0.38 | 0.004 | ArfGAP with SH3 domain, ankyrin repeat and PH domain 2 |
| Pacs2 | 0.38 | 2E-4 | Phosphofurin acidic cluster sorting protein 2 |
| Ncan | 0.42 | 0.008 | Neurocan |
| uc008jhl.1 | 0.42 | 0.010 | |
| Smarcd3 | 0.45 | 0.006 | SWI/SNF Related, Matrix Assoc.Actin Dep.Reg. Chromatin, Subfamily D, Member 3 |
| Tbce | 0.46 | 0.002 | Tubulin-specific chaperone E |
| uc009mzt.1 | 0.47 | 0.008 | |
| Anxa6 | 0.50 | 0.001 | Annexin A6 |

## *Fos* and *Stk11* expression in BLA projection neurons are necessary for memory formation

We next wished to determine whether any of the transcriptional changes in BLApns that correlate with learning are indeed necessary for learning to occur. Since both C-FOS and STK11 protein increase following CTA, we pursued a loss-of-function (LOF) strategy. To restrict LOF to BLApns, we performed conditional deletion by injecting Cre recombinase into mice carrying alleles of *Fos* (*Zhang et al., 2002*) or *Stk11*(*Nakada et al., 2010*) in which key exons are flanked by lox-p sites. In both cases, recombination leads to a functionally null allele and analyses were carried out in homozygous (<sup>f/f</sup>) animals. Cre was delivered by injecting AAV2/5-Camk2α::Cre-GFP into the BLA bilaterally. As expected, this led to GFP expression in excitatory projection neurons, but not in BLA interneurons or adjacent GABAergic neurons in the Central Amygdala (CEA; *Figure 3*, 5). Animals carrying the same genotypes but receiving AAV2/5-Camk2α::GFP served as controls. Injections were performed 10 days prior to analysis to allow time for LOF to occur.

To assess the efficacy of this approach for CTA learning, we first examined the necessity of *Fos* expression. *Fos* is known to contribute to multiple forms of learning and plasticity (*Zhang et al., 2002*; *Tonegawa et al., 2015*) and has previously been implicated in CTA (*Lamprecht and Dudai, 1996*; *Yasoshima et al., 2006*). To first confirm effective Cre-mediated recombination, we tested the ability of viral Cre to prevent widespread C-FOS expression in the BLA immediately following kainic-acid-induced seizures. C-FOS staining performed four hours after seizures revealed strong induction throughout the BLA of control mice injected with the control virus, and diminished C-FOS expression in mice injected with Cre (*Figure 3A,B*). We then tested the effect of *Fos* deletion on CTA in separate animals. Cre-GFP and GFP control AAV's were injected into the BLA of *Fos*<sup>f/f</sup> mice, CTA training occurred 10 days later, and long-term memory was tested after an additional 2 days.

The results show that while both groups could form CTA memory, Cre injected mice showed significantly weaker memory (*Figure 3C–E*). This confirms our ability to manipulate memory by conditional knockout in BLApn neurons. The results refine those of a prior study using antisense injections and germline knockouts (*Yasoshima et al., 2006*).

Next, we used the same strategy to test the necessity of *Stk11* expression in BLApn for CTA memory. BLA of *Stk11*<sup>f/f</sup> mice were injected bilaterally with AAV expressing Cre- GFP, or GFP alone. Ten days later, both groups of mice were trained for CTA and tested after an additional 2 days. The results revealed a near complete loss of learning in the *Stk11* KO mice (*Figure 4A,B*), which exhibited no significant reduction in the amount or ratio of saccharin consumed after training. Control mice exhibited significant reductions consistent with a similar degree of learning to that seen in previous control experiments. The differences in the consumption of saccharin during the test day were not attributable to overall reduced drinking, as both group of animals consumed comparable amounts of water 8 hr later (*Figure 4C*). Taken together, these results indicate that *Stk11* expression in BLApn is essential for CTA memory.

*Stk11* deletion prior to CTA training can potentially alter multiple memory stages including memory formation and retrieval (*Levitan et al., 2016a*; *Levitan et al., 2016b*). Because CTA memory is long-lasting after even a single training session, it is possible to distinguish an effect of *Stk11* deletion on memory formation from an effect on retrieval by performing the deletion figure supplement 1). Both Cre-GFP and GFP injected mice developed CTA (*Figure 4D*) – there was no significant between-group difference in the intensity of memory, assessed from the ratio of saccharin consumed during the test (*Figure 4E*). Since the same deletion produces a profound effect when occurring prior to training, this suggests that deletion of Stk11 from BLApn does not affect the retention and retrieval of CTA memory, provided memory was already formed prior to performing the knockout. This argues that Stk11 is required for CTA memory formation.

The fact that the expression of both *Fos* and *Stk11* in BLApn are needed for normal CTA long-term memory raises the question of their possible interaction. To address this question, we first examined the expression of C-FOS 4 hr following CTA training in *Stk11*<sup>f/f</sup> mice receiving Cre or control virus injection 10 days earlier. There was no difference between the control and knockout groups (t(4)=-0.42; p=0.69, N = 3/group. *Figure 5—figure supplement 1a*). In a complementary experiment, we examined the expression of STK11 4 hr following CTA in Fos<sup>f/f</sup> mice receiving Cre or control virus injection 10 days earlier. We also found no significant difference between the control and knockout groups in this experiment (t(4)=0.028; p=0.84, N = 3/group. *Figure 5—figure supplement*

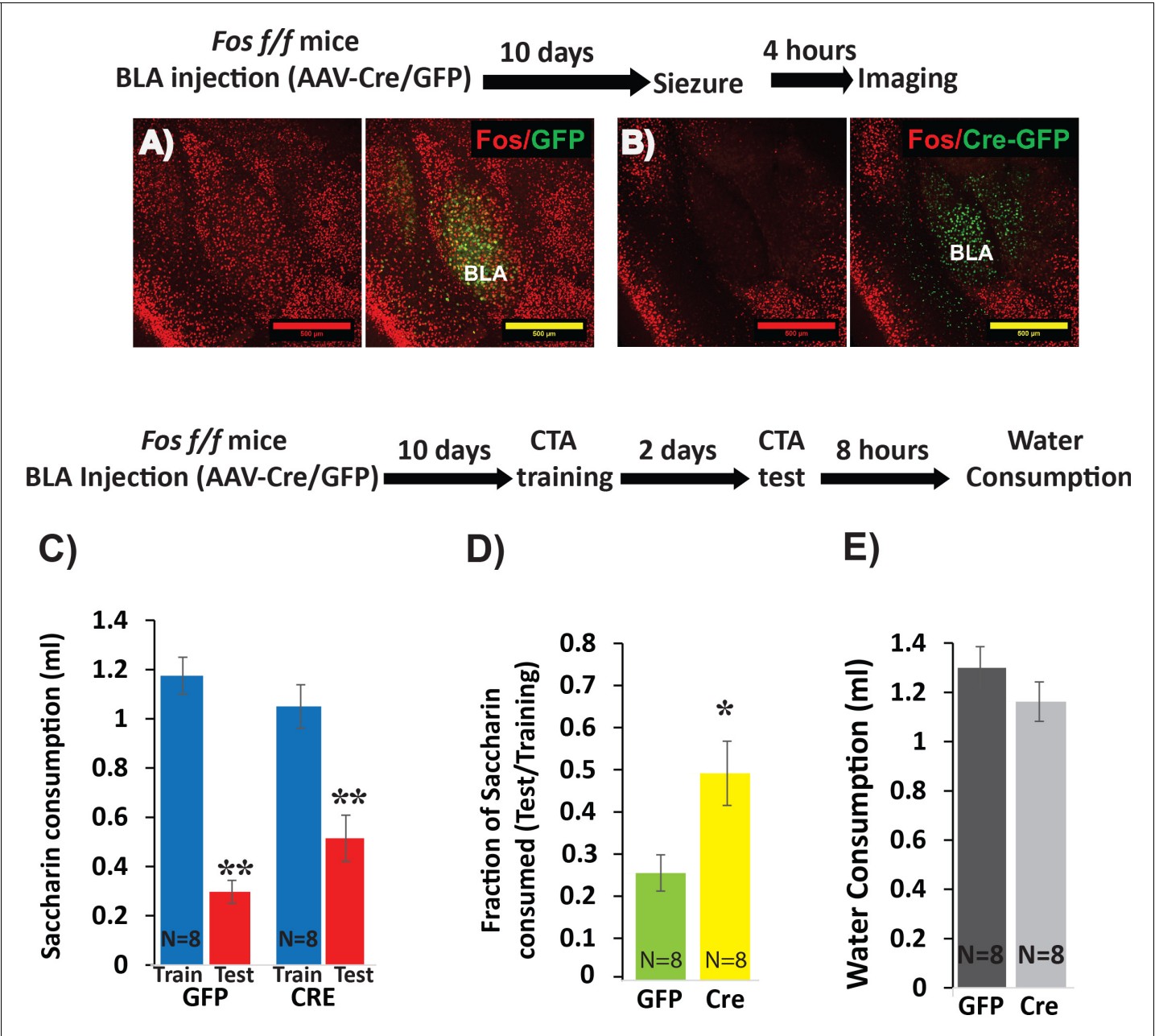

**Figure 3.** Deletion of *Fos* from BLApn reduces the strength of learning. (**A,B**) BLA of *Fos*^f/f mice were infected with viruses expressing Cre-GFP (**B**) or GFP alone (**A**). C-FOS induction was tested 10 days later, 4 hr after onset of seizures in response to kainic acid (20 mg/kg). Cre injected BLA's had reduced C-FOS expression confirming penetrance of the knock-out. (**C–E**) Fos deletion from BLApn attenuates CTA learning. *Fos* f/f mice received Cre and control viruses bilaterally and were trained for CTA 10 days later and then tested after an additional 48 hr. (**C**) Both groups exhibited significant memory reflected by reduced saccharin consumption between training and testing sessions. Mixed ANOVA revealed a significant effect of training (F (1,14) = 154.6, p=5.9×10$^{-9}$), but not of knockout (F(1,14) = 0.15, p=0.70), although there was a significant interaction (F(1,14) = 9.67, p=0.008). Post-hoc analysis revealed that both GFP (N = 8) and Cre (N = 8) group reductions following CTA (test vs. train) were significant (GFP: p=9×10$^{-6}$; Cre: p=3.6×10$^{-4}$) but differences between the other conditions were not (CTA-GFP vs CTA-Cre: p=0.26 and test-GFP vs test-Cre: p=0.065). (**D**) *Fos* deletion from BLApn reduced memory strength measured as the fraction of saccharin consumed (test/training): 25% (GFP) versus 49% (Cre) and this difference in ratios was significant (t(14)=-2.7; p=0.017). (**E**) Reduced saccharin consumption cannot be attributed to overall inhibition of drinking as the amount of water drunk 8 hr later did not differ (p=0.26). *p<0.05; **p<0.01. See also *Figure 3—source datas 1* and *2*.
The online version of this article includes the following source data for figure 3:

**Source data 1.** Saccharin consumption (ml) during CTA training and test.
**Source data 2.** Fraction of saccharin consumed (Test/Training).

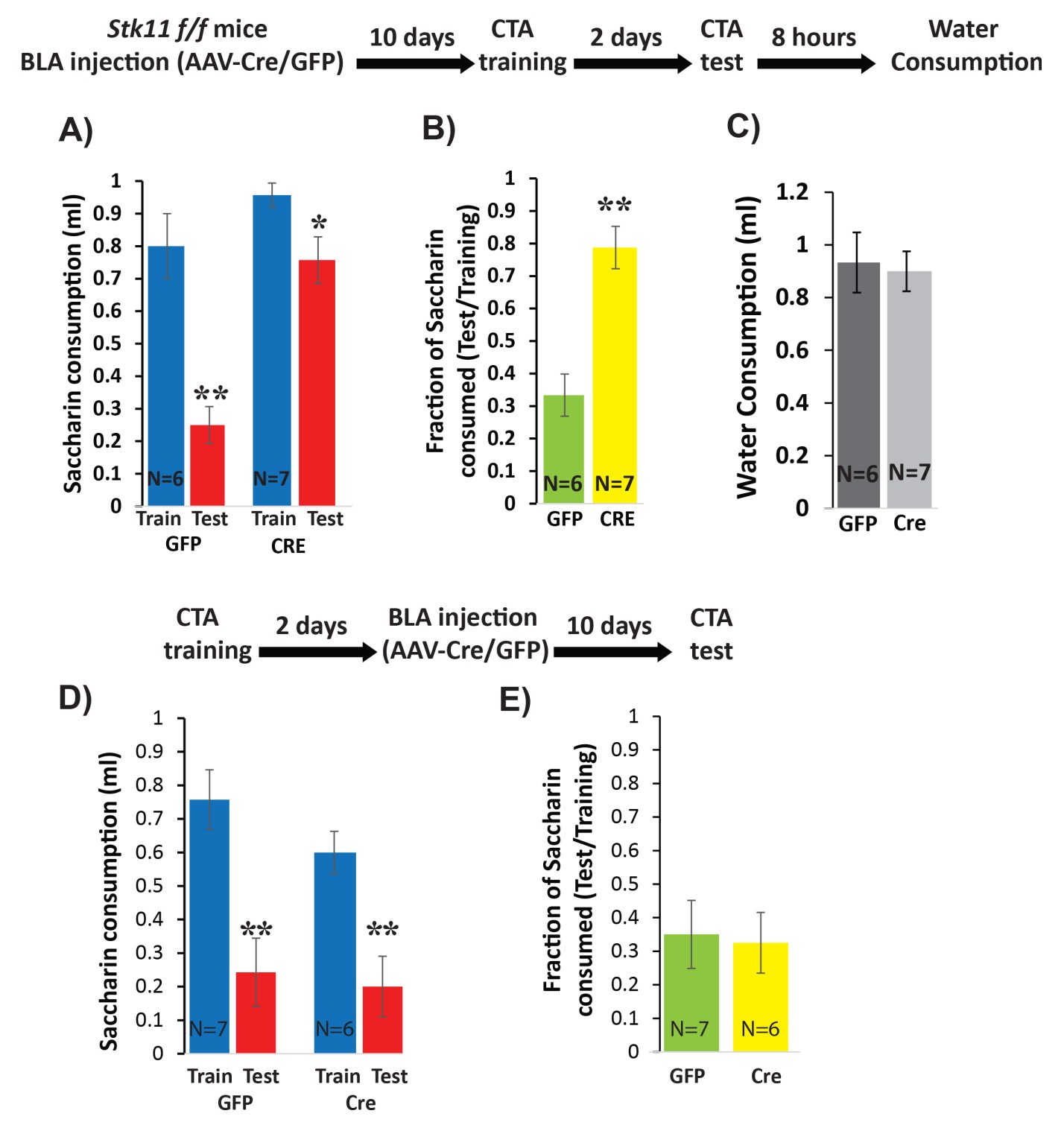

**Figure 4.** Deletion of Stk11 from BLApn impairs CTA learning. (**A**) *Stk11*^f/f mice were infected with Cre or control viruses 10 days before CTA training and were tested 48 hr later. Despite significant reduction in saccharin consumption between testing and training sessions in control mice, *Stk11* KO animals showed almost no learning. Mixed ANOVA revealed significant training and genotype effects (training: F(1,11) = 42.1; p=4.5×10⁻⁵; genotype: F (1,11) = 18.2; p=0.001; as well as a significant interaction: F(1,11) = 9.2, p=0.012). Post-hoc analysis (Bonferroni corrected) revealed that although both GFP (N = 6) and Cre (N = 7) reductions following CTA (test vs. train) were significant (GFP: p=0.003; Cre: 0.013), there was a large and significant difference in saccharin consumption between GFP and Cre groups during the test (GFP: 0.25 ml; Cre: 0.75 ml; p=2.1×10⁻⁴), while differences in saccharin consumption during training were not significant (p=0.15). (**B**) Control animals (GFP) consumed only 33% of the training amount during the

*Figure 4 continued on next page*

*Figure 4 continued*

test, but KO mice (Cre) consumed 78% and this difference was significant (t(11)=-4.91; p=4×10$^{-4}$). (C) Reduced saccharin consumption does not reflect an overall reduction in drinking measured 8 hr later (t(11)=0.25; p=0.87). **p<0.01. (D) *Stk11* deletion after long-term memory formation has no effect on CTA retention. *Stk11* f/f mice received Cre and control viruses 2 days after CTA training and were tested 10 days later. Mixed ANOVA revealed a significant effect of training (F(1,11) = 48.8, p=2.3×10$^{-5}$) but no significant effect of genotype (F(1,11) = 1.7, p=0.23) or interaction (F(1,11) = 0.76, p=0.40). Post-hoc analysis confirmed significant reductions in both groups following CTA (GFP: N = 7, p=0.003. Cre: N = 6, p=0.002). There was no significant difference between GFP and Cre groups during training (p=0.19) or during the test (p=0.64). (E) There was no significant difference in CTA intensity as measured by the fraction of saccharin consumed (t = −0.18, p=0.86). **p<0.01. See also *Figure 4—source datas 1–4* and *Figure 4—figure supplement 1*.

The online version of this article includes the following source data and figure supplement(s) for figure 4:

**Source data 1.** Saccharin consumption (ml) during CTA training and test.
**Source data 2.** Fraction of saccharin consumed (Test/Training).
**Source data 3.** Saccharin consumption (ml) during CTA training and test.
**Source data 4.** Fraction of saccharin consumed (Test/Training).
**Figure supplement 1.** Controls for Stk11 deletion experiments.

*1b*). Thus, the expression of *Stk11* or *Fos* is not dependent upon the expression of the other, and they likely represent signaling pathways that are at least initially separate.

## *Stk11* deletion in BLApn does not impact taste palatability coding in the GC

Since Stk11 deletion must occur prior to CTA training to have an effect on learning, we needed to rule out the possibility that its effect on memory came via disruption of either the responsiveness of BLA neurons to training stimuli, or the output of BLA neurons, which is known to be required for palatability coding within GC.

A more rigorous test of BLA function is to determine whether palatability coding in the GC is intact following knockout, since this is known to depend on intact output from the BLA (*Piette et al., 2012*; *Samuelsen et al., 2012*; *Lin and Reilly, 2012*; *Lin et al., 2018*). If *Stk11* deletion disrupts gustatory activation of BLApn or their output to the GC, palatability coding recorded in the GC should be impaired. To test this, *Stk11* deletion in BLApn was performed as before and 10 days later multi-channel in vivo recordings of GC taste responses were obtained. Recordings were targeted to the ventral part of GC, since BLA projects to these regions (*Figure 5A*; *Maffei et al., 2012*; *Haley et al., 2016*; *Levitan et al., 2019*). Palatability coding was assessed with a battery of four tastes with hedonic values ranging from palatable (sucrose and sodium-chloride) to aversive (citric-acid and quinine; for details see *Levitan et al., 2019*).

*Figure 5* shows the results of GC taste responses following *Stk11* deletions in BLA. *Figure 5B* illustrates the peri-stimulus histogram of a representative GC neuron responding to the taste battery. As observed previously in rats and mice (*Katz et al., 2001*; *Sadacca et al., 2012*; *Levitan et al., 2019*), different aspects of taste processing are encoded in firing rates sequentially. In the first five hundred milliseconds or so post-taste delivery, neurons show different firing rates to different tastes (i.e. reflecting taste identity coding), while later in the responses, differential response rates reflect the hedonic values of tastes (i.e. taste palatability coding). These properties were maintained in the mice studied here: as indicated by the dashed line, the magnitude of the correlation between the neuron's stimulus evoked firing rates and the behaviorally determined palatability ranking rose significantly only after the first half a second following taste delivery. The averaged correlation magnitudes across neurons from Cre and GFP injected mice are shown in *Figure 5C*. Inspection of the figure suggests that BLA *Stk11* deletion had little effects on GC taste palatability coding; the correlations in both GFP- and Cre-injected groups rise around half a second and peaks at about one second after taste delivery. An ANOVA found no significant group differences across each time bin, suggesting that *Stk11* deletion in BLA has little detectable influence on GC taste processing.

Taken together, our molecular and electrophysiological analyses suggest that the memory deficit observed after Stk11 deletion is unlikely to be due to a deficit in basic taste processing. Rather, Stk11 deletion likely impairs memory by affecting the process of memory formation.

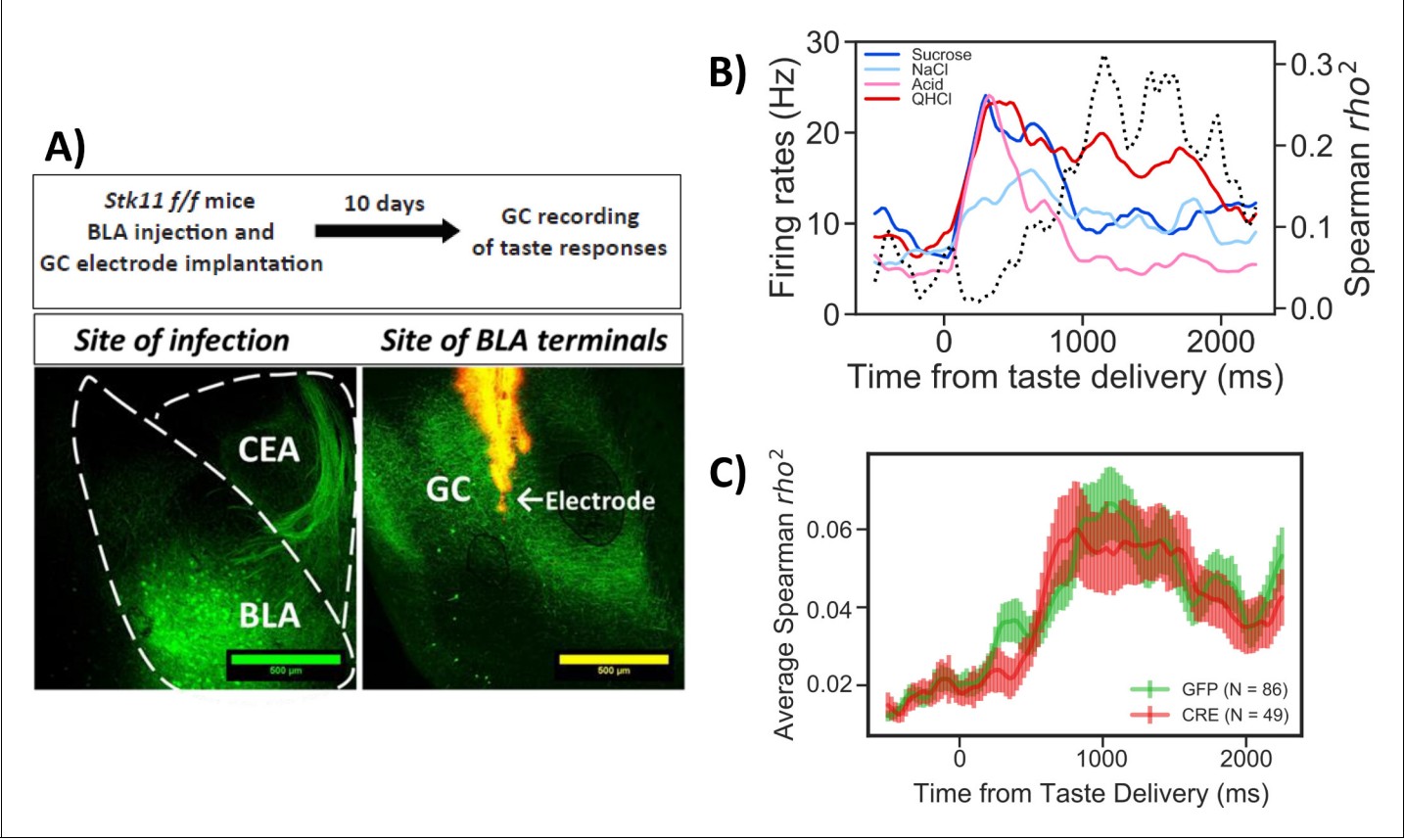

**Figure 5.** *Stk11* deletion in BLApn does not affect taste palatability coding in the GC. (**A**) The BLA of *Stk11*[f/f] mice was infected bilaterally with Cre or control viruses. The ventral GC, where BLA projections terminate (*Haley et al., 2016*) was implanted with a multi-electrode array 10 days later to record GC taste responses to a battery of four tastes differing in their hedonic value, the palatable sucrose and sodium-chloride and the aversive critic acid and quinine (*Levitan et al., 2019*). Images show BLA injection site (left) and labeled BLA terminals in the ventral GC co-localized with the site of dye-labeled electrodes (right). (**B**) PSTHs (colored lines) from a representative GC neuron in a GFP-injected control mouse that responded significantly to all tastes. Dashed line represents the magnitude of the rank correlation between firing rates and behaviorally measured palatability obtained previously in separate experiments (*Levitan et al., 2019*). (**C**) Correlation coefficients averaged across all recorded units in GFP (control) and Cre-injected mice. As revealed in a two-way ANOVA, palatability correlations in both groups rise steeply between 800 and 1000 ms with no significant difference between genotypes ($F_{(1,133)}=0.13$, $p=0.72$) or interaction ($p=0.99$). See also *Figure 5—figure supplement 1*.

The online version of this article includes the following source data and figure supplement(s) for figure 5:

**Figure supplement 1.** The expression of Stk11 or Fos is not dependent upon the expression of the other.

**Figure supplement 1—source data 1.** *Stk11* deletion in BLApn does not change C-FOS expression.

**Figure supplement 1—source data 2.** *Fos* deletion in BLApn does not change C-FOS expression.

### *Stk11* and *Fos* deletion increase the intrinsic excitability of BLApn, while CTA decreases excitability

What cellular mechanisms mediate the effects of *Fos* and *Stk11* on memory formation? Long-term memory is known to be accompanied by changes in both the intrinsic excitability of neurons (*Zhang and Linden, 2003*; *Mozzachiodi and Byrne, 2010*) and in the strength of their synaptic connections. Recent studies have shown that the intrinsic excitability of BLApn can be modulated bi-directionally during reinforcement learning, with positive reinforcement leading to increased and negative reinforcement to decreased intrinsic excitability (*Motanis et al., 2014*).

To determine whether *Stk11* might influence memory formation by altering intrinsic excitability, we compared the excitability of BLApn recorded in ex vivo slices from mutant animals receiving Cre-GFP (*Figure 6*). The results reveal a marked increase in the intrinsic excitability of BLApn following deletion of *Stk11*, relative to GFP-only controls. Cre infected neurons had higher firing rates than

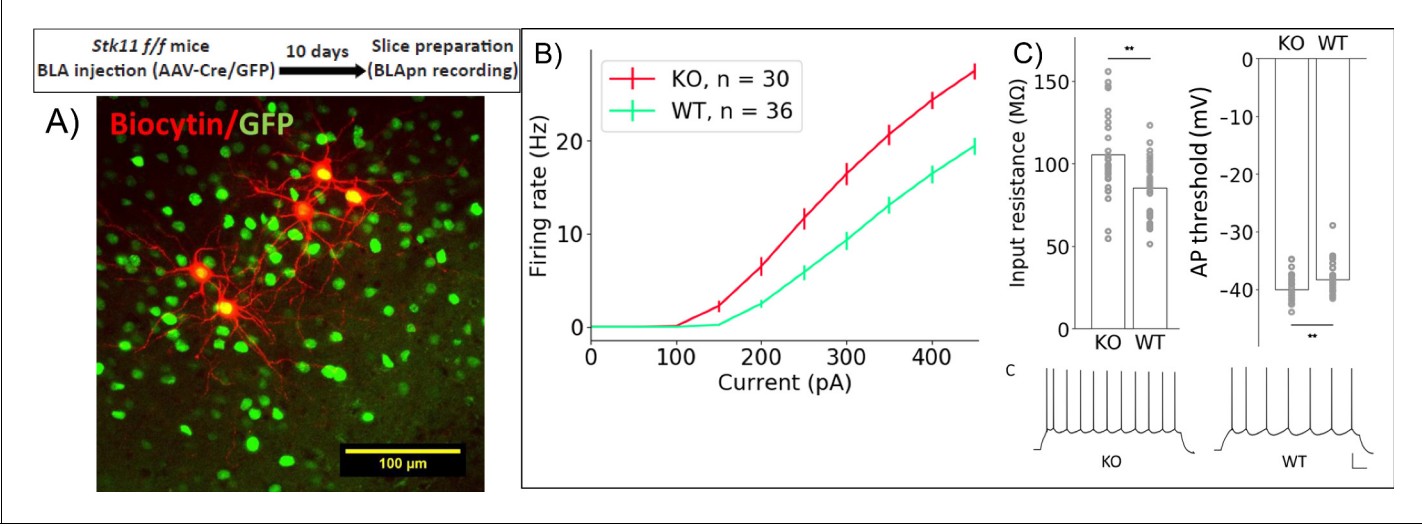

**Figure 6.** *Stk11* deletion in BLApn increases intrinsic excitability. (**A**) Whole-cell recordings obtained from BLApn in ex vivo slices of *Stk11*$^{f/f}$ mice 10 days after injection of Cre or control virus. *Stk11* neurons were targeted based on GFP expression and validated post-hoc based on Biocytin fills. (**B**) Firing rates plotted against input current (F–I). Error bars are SEM. Less current is needed to evoke firing in *Stk11*-KO neurons compared to *Stk11*-WT neurons (interpolated rheobase, $F_{(1,64)} = 10.6$, p=0.0018). Two-way mixed ANOVA revealed significant main effects of knockout ($F_{(1,64)} = 27.48$, p=$2\times10^{-6}$) and current ($F_{(10,640)} = 581.7$, p=$1.2\times10^{-44}$) on firing rate, along with a statistically significant interaction ($F_{(10,640)} = 22.4$, p=$4\times10^{-36}$). Post-hoc pairwise comparisons showed that the firing rates of *Stk11*-KO neurons were significantly greater than those of *Stk11*-WT neurons in response to current injections from 150 to 450 pA (p<0.001). (**C**) *Stk11*-KO neurons have increased input resistance (F = 14.41, p=$3.3\times10^{-4}$) and decreased threshold for generating action potential (F = 7.68, p=0.0073). Traces are sample responses to 300 pA current steps. Scale bar: 100 ms, 20 mV. See also *Table 4*.

GFP neurons for any given amount of current input resulting in a steeper slope of the firing rate vs. current (F-I) curve. Threshold firing was initiated at a lower level of current injection (i.e. the rheobase was lower; *Figure 6C*). This reflected a higher resting input resistance and a slightly lower voltage threshold. Some other electrophysiological properties also differed (see *Table 4*) including sag ratios, action potential amplitudes, and the medium and slow afterhyperpolarizations, while others, such as the degree of firing rate accommodation, spike widths and resting membrane potentials did not. Thus, *Stk11* deletion from BLApn neurons increases overall intrinsic excitability, most likely by affecting multiple biophysical properties of these neurons.

Since *Stk11* and *Fos* deletion both impair CTA learning, we wondered whether *Fos* deletion also increases the intrinsic excitability of BLApn. Analysis of *Fos* KO neurons revealed a similar increase in firing relative to control neurons (N for: WT = 14, KO = 22; *Figure 7A–B*). Two-way ANOVA indicated a significant effect of genotype (in addition to the expected effect of current level). Significant differences in input resistance and action potential threshold were also detected.Next, we asked

**Table 4.** Electrophysiological properties of BLApn: *Stk11* knockout vs. GFP controls

| Group | Statistics | Resting membrane potential (mV) | Access resistance (mΩ) | Input resistance (mΩ) | mAHP (mV) | sAHP (mV) | Action potential amplitude (mV) | Action potential half width (ms) | Action potential threshold (mV) | Sag ratio |
|---|---|---|---|---|---|---|---|---|---|---|
| KO Mice = 11 Neuron = 30 | Mean | −78.23 | 19.68 | 105.33 | 7.44 | 0.92 | 74.41 | 0.80 | −50.01 | 0.13 |
| | S.D. | 3.73 | 2.76 | 24.88 | 1.93 | 0.35 | 6.58 | 0.10 | 2.44 | 0.04 |
| GFP Mice = 11 Neuron = 36 | Mean | −76.96 | 18.02 | 85.41 | 9.71 | 0.64 | 78.05 | 0.79 | −48.30 | 0.15 |
| | S.D. | 2.66 | 3.47 | 17.65 | 1.64 | 0.26 | 4.39 | 0.07 | 2.55 | 0.03 |
| | F | 2.59 | 4.47 | 14.41 | 26.69 | 13.78 | 7.17 | 0.19 | 7.68 | 7.48 |
| | p | 0.112 | 0.0383 | 3.29E-4 | 2.54E-06 | 4.32E-4 | 0.0094 | 0.664 | 0.0073 | 0.008 |

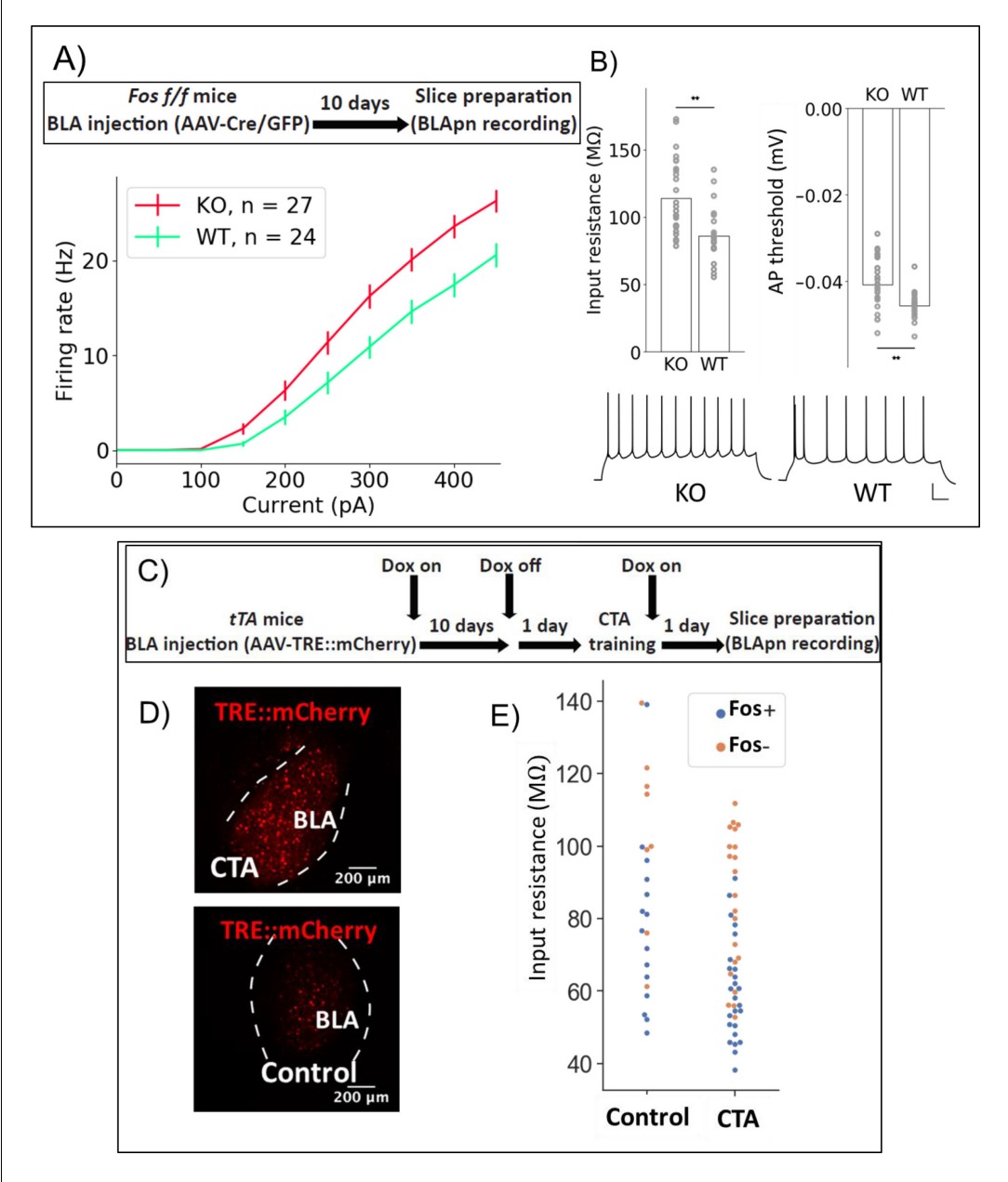

**Figure 7.** *Fos* deletion and CTA have opposing effects on resting input resistance. (**A,B**) Whole-cell patch clamp recordings obtained from BLApn in ex vivo slices of *Fos*^f/f mice 10 days after injection of Cre or control virus. *Fos*-KO neurons exhibit increased firing in response to current injection compared to *Fos*-WT neurons. (**A**) Average frequency-current (FI) curves. Two-way mixed ANOVA revealed a significant main effect of knockout ($F_{(1,49)}$ = 8.79, p=0.005) and current ($F_{(9,441)}$ = 419.4, p=6×10$^{-210}$) on firing rate, along with a statistically significant interaction ($F_{(9,441)}$ = 8.2, p=2.57×10$^{-11}$). Post-hoc pairwise comparisons showed that the firing rates of *Fos*-KO neurons were significantly greater than those of *Fos*-WT neurons in response to current injections from 150 to 450 pA (p<0.05). (**B**) Differences in input resistance ($F_{(1)}$ = 10.62, p=0.0025) and action potential threshold ($F_{(1)}$ = 6.82, p=0.013) between *Fos*-KO and *Fos*-WT neurons were also significant. Traces show sample responses to 250 pA current steps. Scale bar: 100 ms, 20 mV. (**C**) Tet-dependent labeling of *Fos* expressing neurons (*Reijmers et al., 2007*) during CTA training. Fos::tTa mice injected with AAV-TRE::mCherry received food with 40 ppm Doxycycline (Dox) to suppress reporter expression. One day prior to CTA training, Dox was removed. Acute slices were prepared 24 hr following CTA (Saccharin+lithium) or control (Saccharin+saline) training. (**D**) MCherry labeled neurons in the BLA. (**E**) Input resistances of *Fos*$^{+/-}$ neurons in the BLA, following CTA training and taste-only control experiments. Two-way ANOVA reveals that neurons from CTA animals have lower input resistance than those from control animals ($F_{(1,65)}$ = 10.26, p=0.0021) and that *Fos*$^+$ neurons have lower input resistance than *Fos*$^-$ neurons ($F_{(1,65)}$ = 23.64, p=7.5×10$^{-6}$). Post-hoc tests show that among *Fos*$^+$ neurons, those in CTA animals have lower input resistance (p=0.015), while among *Fos*$^-$ neurons, there was no significant difference between CTA and control animals (p=0.1). Differences between *Fos*$^+$ and *Fos*$^-$ neurons did not reach post-hoc significance in either the CTA (p=0.25) or control animals (p=0.12) considered alone. See also *Table 5*.

whether we could detect an effect of learning itself on neuronal excitability in the BLA. A difficulty with this experiment is that learning may have different effects on different populations of BLApn as evidenced by the fact that some neurons increase C-FOS expression following training, while others do not (*Figure 2—figure supplement 2*). In order to separately examine these two populations following training, we made use of the reporter system developed by Reijmers and colleagues, in which elements of the *Fos* promoter are used to drive the tet transactivator (tTA) which can then prolong and amplify expression of a reporter marking *Fos*-activated cells, when Doxycycline (Dox) is absent (*Reijmers et al., 2007*). In this case, we provided the tet-dependent reporter via an AAV (TRE:: mCherry) injected into the BLA 10 days prior to training. Animals were fed Dox until 24 hr prior to training to limit background activation (*Figure 7C*). Control experiments showed that CTA training resulted in a greater number of neurons expressing *Fos* 24 hr after training, compared to home cage controls (*Figure 7D*). Twenty-four hours following CTA training or taste-only control experiments, recordings were obtained from BLApn in acute slices. Two-way ANOVA revealed that BLApn in the CTA animals have lower input resistance than those in control animals and that *Fos*-activated neurons have lower input resistance than unlabeled neurons (*Figure 7E*). Notably, this change is in the opposite direction from that produced by deletions of *Stk11* and *Fos*, two manipulations that impair learning. This suggests that these mutations may interfere with learning by impairing changes in intrinsic excitability that are required for learning to occur.

To more directly test the hypothesis that changes in intrinsic excitability of BLApn play a key role in the formation of CTA memory, we virally expressed the excitatory DREADD (Designer Receptors Exclusively Activated by Designer Drugs) hM3Dq and then transiently increased the intrinsic excitability of BLApn during memory formation. hM3Dq is a Gq-coupled muscarinic receptor containing a modified binding site permitting activation by the synthetic ligand CNO (Clozapine-N-Oxide) but not by endogenous ligands (*Alexander et al., 2009*). Whole-cell patch clamp recordings in acute brain slices from BLApn infected in vivo with AAV-Camk2α::hM3Dq confirmed the ability of CNO to increase their intrinsic excitability (8-figure supplement 1 A-C: higher firing rate for CNO conditions compared to control with 100 pA current input, paired sample t-test, t = −4.73, p=$1\times10^{-4}$; Resting membrane potentials of neurons are significantly higher in CNO conditions than in control (paired sample t-test, t = −8.52, p=$2\times10^{-8}$). In addition, we confirmed that systemic injection of 0.3 mg/kg CNO to mice infected with Camk2α::hM3Dq increased C-FOS expression 60 min later (*Figure 8— figure supplement 1*). Mice infected with Camk2α::hM3Dq/GFP were then trained for CTA 60 min after systemic injection of 0.3 mg/kg CNO and were tested 48 hr later. We found that while both groups of mice developed CTA reflected by significantly reduced saccharin consumption, hM3Dq-injected mice consumed more saccharin during the test then mice receiving the control GFP virus, but did not differ in their consumption of saccharin during training in comparison to GFP injected mice (*Figure 8A*) resulting in significantly stronger CTA (*Figure 8B*). These data show that increasing excitability during CTA training impairs CTA memory formation. Together with our previous data, this suggests that that *Stk11* and *Fos* expression play key roles in CTA long-term memory formation, by modulating the intrinsic excitability of BLApn.

**Table 5.** Electrophysiological properties of BLApn: *Fos* knockout vs. GFP controls

| Group | Statistics | Resting membrane potential (mV) | Access resistance (mΩ) | Input resistance (mΩ) | mAHP (mV) | sAHP (mV) | Action potential amplitude (mV) | Action potential half width (ms) | Action potential threshold (mV) |
|---|---|---|---|---|---|---|---|---|---|
| KO Mice = 5 Neuron = 27 | Mean | −75.43 | 19.63 | 113.91 | 10.34 | 0.40 | 65.87 | 0.72 | −40.79 |
| | S.D. | 5.08 | 4.95 | 27.73 | 4.30 | 0.19 | 8.31 | 0.11 | 5.83 |
| GFP Mice = 5 Neuron = 24 | Mean | −76.50 | 14.35 | 86.01 | 8.95 | 0.38 | 73.30 | 0.74 | −45.62 |
| | S.D. | 3.38 | 5.17 | 19.58 | 2.05 | 0.17 | 5.11 | 0.12 | 3.02 |
| | F | 0.768 | 13.882 | 16.812 | 2.096 | 0.144 | 14.334 | 0.224 | 13.306 |
| | p | 0.385 | 5.04E-04 | 1.55E-04 | 0.154 | 0.706 | 4.18E-04 | 0.638 | 6.40E-04 |

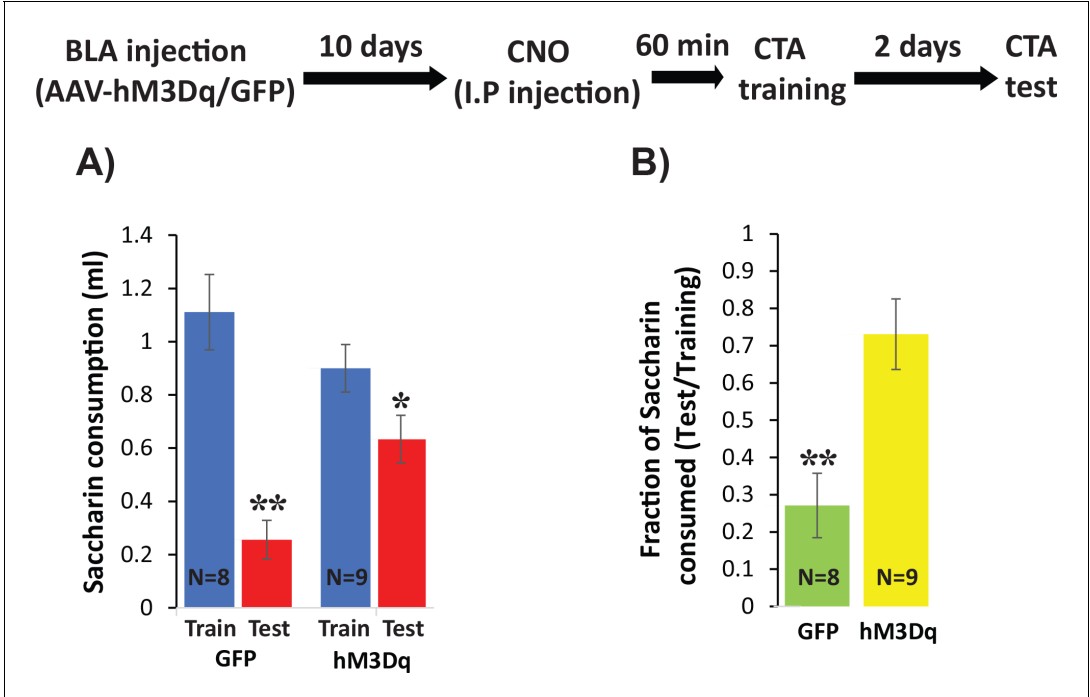

**Figure 8.** Increasing excitability using hM3Dq DREADD in BLApn during CTA training impairs CTA learning. *Stk11^{f/f}* mice were infected with Camk2α:: hM3Dq or Camk2α::GFP control viruses 10 days before CTA training. Mice received systemic CNO (0.3 mg/kg) 60 min before training and were tested for CTA memory 48 hr later. (**A**) Control mice had greater reductions in saccharin consumption between testing and training sessions than hM3Dq mice. Mixed two-way ANOVA revealed a significant training effect and a significant interaction between training and treatment (training: $F_{(1,16)} = 32.6$, $p = 3.2 \times 10^{-5}$; treatment: $F_{(1,16)} = 0.63$, $p = 0.44$; interaction: $F_{(1,16)} = 8.97$, $p = 0.009$). This indicates that the effect of training depends on the treatment condition. Post hoc analysis revealed that although both GFP injected mice (N = 8) and hM3Dq injected mice (N = 9) developed CTA indicated by reduced saccharin consumption (GFP: $p = 0.001$; hM3Dq: $p = 0.019$), hM3Dq injected mice drank significantly more saccharin during the test then GFP mice ($p = 0.005$). There was no significant difference in the consumption of saccharin during training ($p = 0.227$). (**B**) The strength of CTA learning was expressed as the fraction of saccharin consumed between testing and training. GFP controls consumed only 22% during the test, relative to training, but hM3Dq mice consumed 72% and this difference was significant ($t(15) = 3.96$; $p = 0.001$). *$p < 0.05$, **$p < 0.01$. See also *Figure 8—source datas 1–2* and *Figure 8—figure supplement 1*.

The online version of this article includes the following source data and figure supplement(s) for figure 8:

**Source data 1.** Saccharin consumption (ml) during CTA training and test.
**Source data 2.** Fraction of saccharin consumed (Test/Training).
**Figure supplement 1.** HM3Dq DREADD increases BLA activity.

## Discussion

This study is the first to identify a role for Stk11, a master kinase at the top of the AMP-related kinase pathway, in long-term memory. Using CTA as a behavioral paradigm, we first established a causal requirement for transcription in BLApn during establishment of long-term memory, and went on to show that changes in *Stk11* and *Fos* transcription and translation accompany CTA learning. Cell-type-specific conditional knock-out of *Stk11* in BLApn revealed it to be necessary for CTA memory formation but not for retrieval, once memories were established. Slice recordings revealed that *Stk11* modulated the intrinsic excitability of these neurons and further investigations suggested the general importance of excitability changes for memory—deletion of the immediate early gene *Fos* in BLApn altered excitability similarly to *Stk11* deletion, and conversely activation of *Fos* during learning reduced excitability. Finally, a transient chemogenetic increase in BLApn excitability during CTA training also impaired CTA memory. This suggests a model in which *Stk11* and *Fos* expression plays important role in CTA long-term memory formation by modulating the intrinsic excitability of BLApn.

## BLA projection neurons undergo transcription important for CTA learning

It is well established that BLA neurons play a necessary role in CTA learning. Multiple studies confirm that activity in the BLA is required for memory formation and retrieval (*Ferreira et al., 2005*; *Garcia-Delatorre et al., 2014*; *Molero-Chamizo and Rivera-Urbina, 2017*). CTA also requires protein synthesis in the BLA (*Josselyn et al., 2004*), but whether new transcription is also required, and if so, the identities of the required transcripts and the cellular processes they promote were not previously known.

Here, we show that BLA projection neurons (BLApn) undergo transcriptional changes important for CTA memory. Inhibiting transcription during CTA training impairs memory tested 48 hr later. Using cell-type-specific RNA sequencing, we go beyond this simple insight to identify the transcripts that are altered in expression in BLApn four hours after pairing of the conditioned and unconditioned stimuli. For comparison, we also examined changes in transcript levels in pyramidal neurons and parvalbumin-positive interneurons in gustatory cortex. These profiling experiments provide a resource for future investigations of other molecules potentially involved in CTA in BLA and GC.

Perhaps the strongest case for new transcription in BLApn involved in learning can be made for the immediate early gene Fos. It is well known that Fos transcription and translation are activated in the forebrain by a variety of memory paradigms (*Mayford and Reijmers, 2015*), and more specifically by CTA in BLApn (*Uematsu et al., 2015*). The YFP-H neurons studied here include the majority of BLApn in the anterior portion of the nucleus (*Feng et al., 2000*; *Sugino et al., 2006*; *Jasnow et al., 2013*; *McCullough et al., 2016*) and the fact that many of these neurons express Fos protein (*Figure 2—figure supplement 1*) and project to GC (*Figure 5A* and *Haley et al., 2016*) supports the suggestion that they are among the population of BLApn transcriptionally activated by training and participating in the BLA-GC circuit implicated in learning by prior studies (*Grossman et al., 2008*). Since Fos transcript and protein are short-lived (*Spiegel et al., 2014*; *Chowdhury and Caroni, 2018*), the most parsimonious explanation is that training induces new transcription and translation, and that it is these effects that are disrupted by the Fos KO in BLApn (*Figure 3*).

The results of selectively knocking Fos out in BLApn clarify the results of earlier studies in which Fos was manipulated with infusion of antisense oligonucleotides (*Lamprecht and Dudai, 1996*; *Yasoshima et al., 2006*) or via global knockout (which had no effect on CTA; *Yasoshima et al., 2006*). Loss of memory has previously been attributed to inhibition of Fos in central amygdala (*Lamprecht and Dudai, 1996*), or in the amygdala as a whole along with the GC (*Yasoshima et al., 2006*). Our demonstration that knockout restricted to BLApn is sufficient to impair memory does not contradict these earlier studies, but suggests that these projection neurons may be a nexus or bottleneck vital for learning in the circuit.

There is still much to be learned about transcription of *Stk11* in CTA. This transcript is presumably less transient than that of immediate early genes, and may be regulated through a process with complex dynamics. This issue is brought into focus by the fact that, across the time points measured, the transcript in profiled cells was decreased, while in anatomically sub-dissected portions of BLA, STK11 protein was increased. Improved temporal and spatial mapping of transcript and protein levels may help clarify these differences.

Regardless, however, the finding that protein levels of STK11 increase are consistent with the fact that *Stk11* deletion before CTA training impairs CTA memory (*Figure 4*) and with the finding that Stk11 deletion increases BLApn intrinsic excitability (*Figure 6*), which is reduced after CTA training (*Figure 7*) and is needed for CTA memory (*Figure 8*). Opposing changes in STK11 protein and mRNA levels could reflect homeostatic control of this powerful signaling pathway. Translation-dependent feedback of some other proteins is known to reduce mRNA levels through nonsense-mediated decay (*Giorgi et al., 2007*; *Paolantoni et al., 2018*) or other forms of mRNA degradation (*Lin et al., 2020*; *Shoshani and Cleveland, 2020*). Complex feedback control involving changes in RNA export and stability has also been described for MAP kinase signaling (*Sugiura et al., 2003*; *Prieto-Ruiz et al., 2020*), but whether this type of regulation exists for *Stk11* and the AMP-related kinase pathway remains unknown.

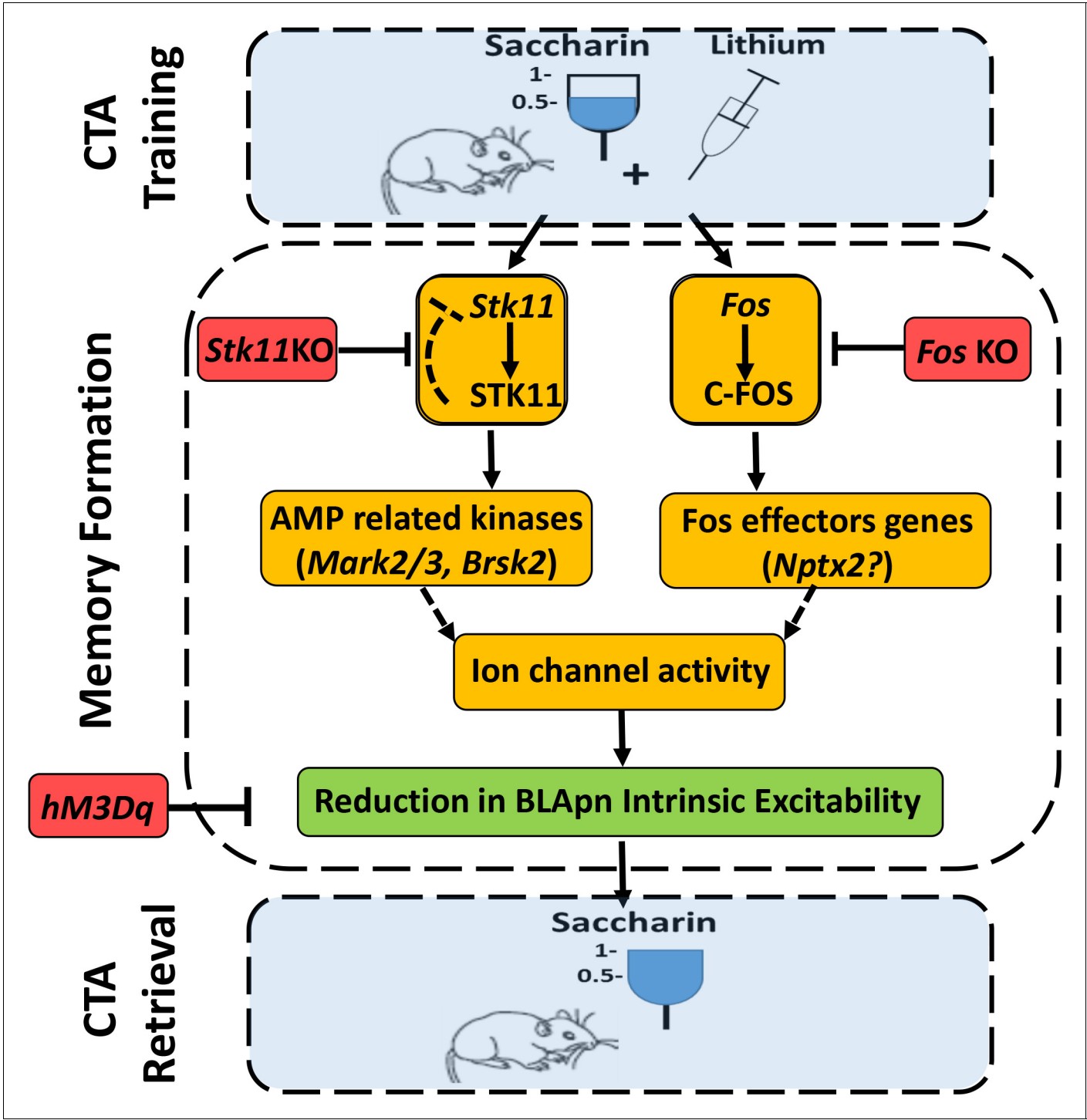

**Figure 9.** An integrated model for the formation of CTA memory. CTA training mice associate consumption of palatable saccharin with gastric malaise (induced by systemic injection of lithium chloride). Memory formation: CTA training increases the activity STK11 and C-FOS in BLA projection neurons (BLApn) via initially separate signaling pathways. CTA increases STK11 protein but reduces its mRNA, possibly through feedback control. STK11 activation results in downstream phosphorylation and activation of AMP related kinases, likely including Mark2/3, Brsk2, and Sik3, which are abundant in BLApn. C-FOS, an activity dependent transcription factor, induces the transcription of effector genes such as *Nptx2* which increases following CTA training and is known to bind not only ligand-gated receptors, but also voltage-gated ion channels. The two pathways reduce BLApn intrinsic excitability through unknown mechanism(s). CTA retrieval: Reduced BLApn intrinsic excitability plays a key role in CTA memory formation, reducing saccharin consumption in test trials. Blue-behavioral level, Orange-molecular level, Green-cellular level, Red-conditions impairing memory. Dashed line arrows are hypothesized connections.

## Necessity of Stk11 implicates the AMP-related kinase pathway in learning

Prior studies of CTA and other forms of aversive learning in the BLA have implicated a number of kinases: including those in the cAMP-dependent protein kinase, protein kinase C, extracellular signal-regulated, and mitogen-activated protein kinase pathways (*Johansen et al., 2011*; *Adaikkan and Rosenblum, 2012*). Each of these also have well-established roles in other forms of forebrain learning and plasticity (*Alberini, 2009*). Here, we reveal the likely involvement of another kinase cascade, well studied in the contexts of cell growth, metabolism, cancer and polarity (*Shackelford and Shaw, 2009*) but hitherto unstudied in the context of learning and memory. That this pathway should have a role in learning is perhaps not shocking given the ubiquity of its previously demonstrated roles in 1) axonal development *Barnes et al., 2007*; *Shelly et al., 2007*; 2) synaptic remodeling during aging *Samuel et al., 2014*; 3) regulation of presynaptic neurotransmission *Kwon et al., 2016*; and perhaps most tellingly 4) regulation of glucose metabolism, feeding and obesity through actions in multiple tissues including hypothalamus (*Xi et al., 2018*; *Fei-Wang et al., 2012*; *Claret et al., 2011*). Given the involvement of hypothalamus in coding of taste palatability, and the connectivity between hypothalamus and gustatory cortex (*Li et al., 2013*), it is tempting to speculate that the role of Stk11 signaling pathways in feeding may be functionally related to its role in gustatory learning. Further studies will be needed to distinguish whether the involvement of Stk11 in memory is specific to forms of learning regulating consumption, and whether its role in CTA learning is confined to the basolateral amygdala.

STK11 is a master kinase that regulates the activity of 13 downstream AMP-related kinases with diverse roles (*Lizcano et al., 2004*). PRKAA1/2 (also known as AMPK) is crucial for metabolic regulation during altered levels of nutrients and intracellular energy. BRSK and MARK regulate cell polarity during development (*Barnes et al., 2007*; *Shackelford and Shaw, 2009*). We found that, while *Prkaa1/2* and several other downstream kinase transcripts have low levels of expression in BLApn, others, including *Mark2* and *3*, are expressed at higher levels (*Figure 2—figure supplement 3*). Furthermore, changes in *Mark2* and *Stk11* expression were correlated during CTA learning raising the possibility that both kinases in the pathway contribute to this novel signaling role in learning and memory.

Deletion of Stk11 prior to training profoundly impaired memory, but deletion two days after training—when memory formation and consolidation have already occurred (*Alberini, 2009*; *Gal-Ben-Ari et al., 2012*; *Levitan et al., 2016b*) —did not. This suggests that Stk11 expression in BLApn promotes memory formation, rather than memory maintenance or retrieval. It is clear that Stk11 deletion left much of the machinery of taste processing and learning intact, however. Activation of Fos by training in the BLA was not impaired after Stk11 deletion, implying that at least the initial stages of transcriptional activation associated with learning are intact. Also left intact was the ability of the BLA to convey palatability information to the GC. Prior studies have shown that silencing of BLA neurons (*Piette et al., 2012*) impair palatability coding during the late phase of GC gustatory responses. We found that these responses were still present in GC following knockout, implying that this critical function of the BLA for CTA learning remained intact.

## Intrinsic excitability of BLApn as a candidate mechanism for Fos and Stk11's effects on CTA memory

Learning paradigms that support synaptic plasticity also frequently induce changes in neuronal excitability, and such plasticity of intrinsic excitability has long been known to accompany classical conditioning in the neocortex, olfactory cortex, hippocampus, amygdala and cerebellum (for reviews see *Zhang and Linden, 2003*; *Frick and Johnston, 2005*; *Mozzachiodi and Byrne, 2010*; *Titley et al., 2017*; *Debanne et al., 2019*).

Although we cannot rule out the possibility that *Stk11* or *Fos* expression in BLApn promotes other forms of plasticity (such as changes in synaptic strength as was suggested in *Yassin et al., 2010*) for *Fos* expression in neurons), our work provide evidence that in BLApn both genes promote changes in intrinsic excitability important for CTA memory. Two genetic manipulations of BLApn that impair learning (deletion of *Stk11* and *Fos*) also increase intrinsic excitability, whereas BLApn involved in normal conditioning appear to undergo the opposite change. Moreover, transiently increasing BLApn intrinsic excitability during training impaired CTA memory, suggesting a model in which

*Stk11* and *Fos* promote changes in BLApn intrinsic excitability necessary for CTA memory. This suggests an important role in learning for changes in excitability, and begs the question of mechanism. Although there are likely several such mechanisms, the increase in excitability partly reflects an increase in the resting input resistance and a corresponding decrease in the threshold current needed to evoke firing. It is worth noting that most prior studies of intrinsic plasticity have reported increases in excitability with learning in various brain regions and different animal models (*Zhang and Linden, 2003*; *Mozzachiodi and Byrne, 2010*; *Yiu et al., 2014*; *Whitaker et al., 2017*; *Chandra and Barkai, 2018*; *Pignatelli et al., 2019*). Our results are not without precedent, however; and are similar in polarity to those found in BLApn following olfactory fear conditioning, another form of negative reinforcement learning (*Motanis et al., 2014*). Decreased excitability of Fos-activated neurons was also found using an earlier reporter of Fos promoter activation (*Yassin et al., 2010*).

Although both *Stk11* and *Fos* expression influence BLApn intrinsic excitability, several observations argue this reflects convergence of two at least partially separate signaling pathways in the BLApn neurons. First, deletion of *Stk11* did not block the ability of training to elicit C-FOS expression, and deletion of *Fos* did not block STK11 expression. Additionally, *Stk11* deletion in BLApn had a stronger effect on CTA memory and on intrinsic excitability then deletion of *Fos*. However, our data do not exclude the possibility for synergy between the STK11 and C-FOS signaling pathways. Each signaling pathway could influence excitability by modulating different ion-channels. Alternatively, both signaling pathways may converge on the same ion-channels but influence complementary processes such as expression (*Kourrich, 2005*), post-translational modification (*Vernon and Irvine, 2016*) or trafficking (*Steele et al., 2007*). If future studies identify the specific ion channels targeted by STK11 and C-FOS related signaling, these questions of synergy and convergence can be more precisely addressed. Our data do not exclude the possibility that the neurons expressing *Fos* represent a sub-population BLApn expressing *Stk11*. Future studies will be aimed to determine with a much finer resolution the distribution of these genes among the population of BLApn.

In conclusion, our data suggest the following model of CTA (*Figure 9*). Training induces the activation of STK11 and C-FOS by increasing their protein levels in BLApn. STK11 activation results in downstream activation (through phosphorylation) of AMP-related kinases, most likely MARK2/3, BRSK2, SIK3, which are abundant in BLApn. C-FOS, an activity dependent transcription factor, induces the transcription of effector genes such as *Nptx2* which is increased following CTA training. NPTX2 may modulate intrinsic excitability through its ability to bind ion-channels (*Duzhyy et al., 2005*; *Chang et al., 2010*). The two pathways jointly reduce the intrinsic excitability of BLApn, which plays a key role in gating the formation of CTA memory.

## Materials and methods

### Subjects

Male and Female mice (total of 209 animals) were used for behavior at age 60–80 days, or for electrophysiology at 25–35 days. Strains: wild-type; WT (C57BL/6J), YFP-H (B6.Cg-Tg(Thy1-YFP)HJrs/J), *Stk11*[f/f] (B6(Cg)-Stk11tm1.1Sjm/J, Lkb1fl), *Fos*[f/f] (B6;129-Fostm1Mxu/Mmjax), *Fos-tTA* (B6.Cg-Tg(Fos-tTA,Fos-EGFP*)1Mmay/J) all purchased from Jackson Laboratories (Bar Harbor, ME). Mice were placed on a 12 hr light-dark cycle, and given ad libitum access to food and water except during training, at which time water access was restricted, while food remained available ad libitum (note that animals reliably consume less food when thirsty). All procedures were approved by the Brandeis University Institutional Animal Care and Use Committee (IACUC, Protocol #20002) in accordance with NIH guidelines.

### Surgery

BLA cannulation for RNA synthesis inhibition experiments

WT Mice were anesthetized via ip injections of 100 µg ketamine, 12.5 µg xylazine, 2.5 µg acepromazine per gram (KXA). Guide cannulae (23-gauge, 10 mm length) were implanted above the BLA (mm from bregma, AP = −1.4, DV = 4.2, ML=±3.4; *Allen Institute for Brain Science, 2019*) and stabilized using Vetbond and dental acrylic. Stainless steel stylets (30-gage, 10 mm) were inserted into the guide cannula to ensure patency. Mice received postsurgical metacam (5 µg/g), penicillin (1500

Units/g) and saline (5% bodyweight) per day for three days and recovered a total of 7 days prior to training. Twenty minutes prior to CTA training, mice were infused with either 50 ng of actinomycin-D or vehicle control (PBS) bilaterally (in 1 µl over 2 min) via infusion cannulae extending 0.5 mm below the guide cannulae to reach the BLA. Each cannula was connected to a 10 µl Hamilton syringe on a syringe pump (Harvard Apparatus, MA, USA).

## BLA viral infection

 Stk11[f/f], Fos[f/f] and Fos-tTA mice were anesthetized with KXA. The skull was exposed, cleaned, and bilateral craniotomies were made at stereotactic coordinates (AP = −1.4, ML=±3.4). BLA were injected bilaterally with AAV2/5-Camk2α::Cre-GFP or AAV2/5-Camk2α::GFP (UNC, vector core) for Stk11[f/f] and Fos[f/f] mice, Camk2α::hM3Dq-mCherry or AAV2/5-Camk2α::GFP (UNC, vector core) for Stk11[f/f] mice and AAV2/5-TRE::mCherry for Fos-tTa mice, 10 days prior to CTA training using sterile glass micropipettes (10–20 µm diameter) attached to a partially automated microinjection device (Nanoject III Microinjector, Drummond Scientific). The micropipettes were lowered to 4.3 mm and 4.6 mm from the dura to reach the BLA. At each depth, virus (200 nl) was delivered via 10 pulses of 20 delivered every 10 s, with 10 min between each injection. Postsurgical treatment and recovery were as above.

## Conditioned taste aversion (CTA)

Mice were housed individually with free access to food and maintained on a 23.0 hr water deprivation schedule for the duration of training and experimentation. Three days prior to CTA training, water bottles were removed from the cages and water was given twice a day (10 am and six pm) for a duration of 30 min. On the day of CTA mice were given 30 min to consume 0.5% saccharin, which was followed by intraperitoneal (I.P) injection of lithium-chloride (0.15M, 2% of body weight, unless indicated differently) 30 min later. Taste control groups received I.P injection of saline (0.9% sodium chloride) instead of lithium and lithium control group received lithium injection alone 24 hr following Saccharin consumption. CTA testing: Mice were kept on watering schedule twice a day and 48 hr after CTA training mice received CTA testing which consisted of 30 min consumption of 0.5% saccharin. 8 hr following testing mice were given 30 min water consumption.

## Seizure induction

Fos[f/f] mice were housed individually with free access to water and food and received BLA viral infection with Cre and control viruses as described above. After 10 days, mice were injected I.P. with 20 mg/kg kainic acid in PBS. Four hours after injection, mice were perfused for Fos immunohistochemistry.

## CNO treatment

CNO (sigma) was diluted in saline to 1 mg/ml. CNO was systemically injected at a dose of 0.3 mg/kg 1 hr prior to CTA training. We chose this dose as we notice that higher concentration (2 mg/kg) causes seizures as was reported (Alexander et al., 2009) and reduction of saccharin consumption. A dose of 0.3 mg/kg did not cause any effect on basal drinking behavior and could strongly activate infected neurons (Figure 8—figure supplement 1), as reported previously by Alexsander et al.

## Immunohistochemistry

Mice were deeply anesthetized with an overdose of KXA and perfused transcardially with phosphate buffered solution (PBS) followed by 4% paraformaldehyde (PFA). Brains were post-fixed in PFA for 1–2 day, and coronal brain slices (60 µm) containing the BLA (−1 mm to −2.5 mm anterior-poterior axis) were sectioned on a vibratome. Slices were rinsed with PBS and incubated in a blocking solution (PBS/.3%TritonX-100/5% Bovine serum albumin) for 12–24 hr at 4˚C. Blocking solution was removed and replaced with the primary antibody solution which consists of 1:100 C-FOS polyclonal rabbit IgG (SC-52G; Santa Cruz Biotechnology) for 24 hr at 4˚C. After incubation, slices were rinsed using a PBS/.3% Triton X-100 solution followed by the secondary antibody incubation of 1:500 C-FOS Alexa Flour 546 Goat-Anti-Rabbit IgG (H+L) (Life Technologies) and 5% natural goat serum for 12–24 hr at 4˚C. Sections were then rinsed 5–6 times over 90 min (1XPBS/.3% Triton X-100), counterstained with DAPI, mounted with antifade mounting medium (Vectashield), and viewed by

confocal fluorescence microscopy (Leica Sp5 Spectral confocal microscope/Resonant Scanner). Imaging and quantification were performed blind to experimental group.

## C-FOS quantification and analysis

To minimize systematic bias, C-FOS counts were performed blind and semi-automatically, using FiJi (University of Wisconsin-Madison; *Schindelin et al., 2012*). Eight-bit images were binarized and particles smaller than 10 $\mu m^2$ were rejected. Each cell count was from a separate animal and was the average of counts from six sections through the anterior, middle and posterior regions of the BLA of both hemispheres.

## RNA sequencing experiment

RNA sequencing was performed on $YFP^+$ BLApn harvested from male YFP-H mouse line (*Feng et al., 2000*; *Sugino et al., 2006*; *Jasnow et al., 2013*; *McCullough et al., 2016*) which expresses YFP under the Thy1 promoter in the majority of excitatory projection neurons located in the anterior part of the nucleus. The mice underwent CTA training or taste-only controls (n = 4/ group) and 4 hr following training were subjected to manual cell-sorting, performed as previously described (*Sugino et al., 2006*; *Hempel et al., 2007*; *Shima et al., 2016*) by dissociating 150–200 fluorescently labeled neurons in 300-μm-thick brain slices and manually purifying them through multiple transfer dishes with the aid of a pipette viewed under a fluorescence dissection microscope. Total RNA was extracted from sorted cells using Pico-pure RNA isolation kit (Thermo fisher). Amplified cDNA libraries are prepared from isolated, fragmented RNA using the NuGen Ovation RNAseq V.2 kit (NuGEN, San Carlos, CA) and followed by purification using the Beckman coulter Genomic's Agencourt RNA Clean XP kit and Zymo DNA Clean and Concentrator. Sequencing adaptors are ligated per Illumina protocols and 50 bp single-ended reads are obtained from Illumina Hi-Seq machine. Libraries sequenced usually results in 25–30 million unique reads using eightfold multiplexing.

## Analysis

Reads are trimmed and then aligned to the mouse genome using TopHat (*Trapnell et al., 2012*). Sam files are converted to binary format using Samtools and visualized at the sequence level using IGV. Script written in python and R statistical package are used to convert unique reads to gene expression values and to filter genes by relative expression and statistical significance. Sequencing data was uploaded to GEO (accession number: GSE138522).

## qPCR validation

The brains of a separate group of YFP-H mice receiving CTA (taste+lithium, N = 4) or taste control (taste+saline, N = 4) were harvested 4 hr following the end of the training and subjected to fax sorting. RNA was extracted using pico-pure kit and reverse transcribed to cDNA using iScript cDNA synthesis kit. qPCR was performed on Rotor-Gene qPCR machine using PCR master mix and transcript-specific sets of primers for *Fos*, *Stk11* as target genes and Snap47 as loading control.

## Acute slice electrophysiology

Ten days after virus injection, acute brain slices were prepared from P28-35 mice. Animals were deeply anesthetized with KXA and transcardially perfused with ice-cold oxygenated cutting solution containing (in mM): 10 N-methyl-D-glucamine (NMDG), 3 KCl, 1 NaH2PO4, 25 NaHCO3, 20 HEPES, 2 Thiourea, 3 Sodium Pyruvate, 12 N-acetyl-L-cysteine, 6 MgCl2, 0.5 CaCl2, 5 Sodium Ascorbate, 10 Glucose (pH: 7.25–7.4, adjusted using HCl). 300 μm coronal slices containing the BLA were cut on a vibratome (Leica), and then recovered for 15 min at 33°C and for 15 min at room temperature in oxygenated recovery solution containing (in mM): 74 NaCl, 3 KCl, 1 $NaH_2PO_4$, 25 $NaHCO_3$, 6 $MgCl_2$, 0.5 $CaCl_2$, 5 Sodium Ascorbate, 75 Sucrose, 10 Glucose, followed by at least another 1 hr at room temperature in oxygenated ACSF containing (in mM): 126 NaCl, 3 KCl, 1 $NaH_2PO_4$, 25 $NaHCO_3$, 2 $MgCl_2$, 2 $CaCl_2$, 10 Glucose. During recordings, slices were perfused with oxygenated 34–35°C ASCF. Target neurons in BLA were identified based on the presence of viral GFP reporter. ACSF included 35 μM d,l-2-amino-5-phosphonovaleric acid (APV) and 20 μM 6,7-dinitroquinoxaline-2,3-dione (DNQX) to block ionotropic glutamate receptors, and 50 μM picrotoxin to block ionotropic

GABA receptors. Whole-cell recording pipettes (6–8 MΩ) were filled with internal solution containing (in mM): 100 K-gluconate, 20 KCl, 10 HEPES, 4 Mg-ATP, 0.3 Na-GTP, 10 Na-phosphocreatine, and 0.1% biocytin. Recordings were amplified (Multiclamp 700B, Molecular Devices) and digitized at 10 kHz using a National Instruments Board under control of IGOR Pro (WaveMetrics). Resting membrane potentials were adjusted to −70 mV and steady state series resistance was compensated. Series resistance and input resistance were calculated using −5 mV (voltage clamp) or 25 pA (current clamp) seal tests before each trial of recording. Measurements of input resistance in Fos reporter labeled neurons (*Figure 7*) were measured in voltage clamp, all other recordings were performed in current clamp. The calculated liquid junction potential (−10 mV) was compensated post hoc. Neurons with high series resistance (>30 MΩ current clamp) or membrane potentials that changed by >10 mV were excluded. Hyperpolarization activated sag was measured from responses to −100 pA current steps. Action potential (AP) threshold, amplitude, afterhyperpolarization (AHP) and full width at half-height were averaged from the 5th-10th APs in trials with 10 to 20 Hz firing rates. AP threshold is the membrane potential at which the slope first exceeds 10 V/s, and AP amplitude was measured relative to threshold. Sag ratio is defined as the fraction by which the membrane potential depolarized at steady-state from its maximum hyperpolarization during a −100 pA current step. Medium AHP was measured as the peak hyperpolarization after the APs mentioned above relative to threshold. The slow AHP was measured from the peak hyperpolarization following positive current steps generating 10–20 Hz firing.

## In vivo recording of GC taste responses

### Surgery

*Stk11*^f/f^ mice were anesthetized and prepared for stereotaxic surgery and viral infection as above. Each mouse was also implanted bilaterally with multi-channel electrode bundles (16 formvar-coated, 25 µm diameter nichrome wires) in GC (Distance from Bregma: AP=+1.2 mm; ML = ± 3 mm; DV of −2.25 mm from the *pia mater*) and a single intraoral cannula (IOC; flexible plastic tubing) was inserted into the cheek to allow controlled delivery of taste stimuli. 24 hr before recording sessions began electrode bundles were then further lowered by 0.75–1.00 mm to reach ventral GC (see *Figure 5A*).

### IOC fluid delivery protocol

Experiments began with three days of habituation to the recording setup and to receiving liquid through the IOC. Sixty 15 µl aliquots (hereafter, 'trials') of water were delivered across 30 min. To ensure adequate hydration, mice were given two 30 min period of access to additional water.

On the following day, recording commenced and water trials were replaced with four different taste stimuli: sweet (0.2 M sucrose), salty (0.1 M sodium chloride), sour (0.02 M citric acid), and bitter (0.001 M quinine). A total of 15 trials were delivered for each taste in random order. These tastes and concentrations were chosen because they provided a broad range of hedonic values for palatability assessment. Fluid delivery through a nitrogen-pressurized system of polyethylene tubes was controlled by solenoid valves via a Raspberry Pi computer (construction details and code available on request from [https://github.com/narendramukherjee/blech_clust] *Mukherjee et al., 2020*; copy archived at https://github.com/elifesciences-publications/blech_clust]).

### Taste palatability coding

To determine whether a neuron displays palatability activity, we performed a moving window analysis (window size: 250 ms; step size: 25 ms) to trace the dynamics of taste processing in GC. For each time window, we calculated a Spearman product-moment correlation between the ranked firing rates to each taste and the palatability rankings obtained previously in separate experiments (*Levitan et al., 2019*).

## Research resource identifiers (RRIDs)

### Software

TopHat, RRID:SCR_013035;
Samtools, RRID:SCR_002105; IGV, RRID:SCR_011793; IGOR Pro, RRID:SCR_000325; Fiji, RRID: SCR_002285.

## Antibodies

FOS-Santa Cruz Biotechnology Cat# sc-52, RRID:AB_2106783.
FOS- Santa Cruz Biotechnology Cat# sc-166940, RRID:AB_10609634.
STK11-Millipore Cat# 09–495, RRID:AB_10807178;
STK11-Sigma-Aldrich Cat# SAB4502888 RRID:AB_10746328
Alexa Fluor 647 Goat Anti-Rabbit IgG (H+L)- Thermo Fisher Scientific Cat# A-21245, RRID:AB_2535813.

## Viruses

AAV-CaMKIIa-GFP-Cre; AAV-CaMKIIa-EYFP and AAV-CamkIIa-hM3Dq-mCherry, (AAV5)-UNC Joint Vector Laboratories, RRID:SCR_002448.

## Mice strains

Wild type-RRID:IMSR_JAX:000664; YFP-H (B6.Cg-Tg(Thy1-YFP)HJrs/J)- RRID:IMSR_JAX:003782;
$Stk11^{f/f}$ (B6(Cg)-Stk11tm1.1Sjm/J, Lkb1fl)- RRID:IMSR_JAX:014143.
$Fos^{f/f}$ (B6;129-Fostm1Mxu/Mmjax)- RRID:MMRRC_037115-JAX.
$Fos$-tTA (B6.Cg-Tg(Fos-tTA,Fos-EGFP*)1Mmay/J)- RRID:IMSR_JAX:018306.

## Statistical analysis

The results are expressed as means ± s.e.m unless otherwise stated. All effects were evaluated using either paired t-test or one- or two-way ANOVA test with post hoc t-tests corrected (Bonferroni) for all possible pair-wise comparisons. Two-way mixed ANOVAs were performed using a mixture of between-subjects and repeated-measures factors depending on the design (e.g. repeated measures for multiple current injections to the same neurons in *Figures 6* and *7*). No power analysis was performed and numbers of replicates performed were the minimum needed to demonstrate reproducibility, consistent with practices in similar published studies.

# Additional information

## Competing interests

Sacha B Nelson: Reviewing editor, *eLife*. The other authors declare that no competing interests exist.

## Funding

| Funder | Grant reference number | Author |
| --- | --- | --- |
| National Institute on Deafness and Other Communication Disorders | DC006666 | Donald B Katz |
| National Institute of Neurological Disorders and Stroke | NS109916 | Sacha B Nelson |

The funders had no role in study design, data collection and interpretation, or the decision to submit the work for publication.

## Author contributions

David Levitan, Conceptualization, Investigation, Writing - original draft, Writing - review and editing; Chenghao Liu, Investigation, Writing - original draft, Writing - review and editing; Tracy Yang, Investigation, Writing - original draft; Yasuyuki Shima, Jian-You Lin, Joseph Wachutka, Yasmin Marrero,

Ramin Ali Marandi Ghoddousi, Eduardo da Veiga Beltrame, Troy A Richter, Investigation; Donald B Katz, Sacha B Nelson, Conceptualization, Writing - original draft, Writing - review and editing

### Author ORCIDs

David Levitan 🔟 https://orcid.org/0000-0002-0332-3708
Tracy Yang 🔟 http://orcid.org/0000-0003-2437-9257
Donald B Katz 🔟 http://orcid.org/0000-0002-8444-6063
Sacha B Nelson 🔟 https://orcid.org/0000-0002-0108-8599

### Ethics

Animal experimentation: All procedures were approved by the Brandeis University Institutional Animal Care and Use Committee (IACUC, Protocol #20002) in accordance with NIH guidelines.

### Decision letter and Author response

Decision letter https://doi.org/10.7554/eLife.61036.sa1
Author response https://doi.org/10.7554/eLife.61036.sa2

## Additional files

### Supplementary files

• Transparent reporting form

### Data availability

Sequencing data was uploaded to GEO (accession number: GSE138522).

The following dataset was generated:

| Author(s) | Year | Dataset title | Dataset URL | Database and Identifier |
|---|---|---|---|---|
| Levitan D, Nelson SB | 2019 | Cell type-specific RNA-seq to profile transcriptional changes in sorted BLA projection neurons during conditioned taste aversion learning | https://www.ncbi.nlm.nih.gov/geo/query/acc.cgi?acc=GSE138522 | NCBI Gene Expression Omnibus, GSE138522 |

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
