## [Decision Letter]

**Acceptance summary:**

The underlying cellular and molecular mechanisms of long-lasting taste aversive memory are not well elucidated. This manuscript tackles this issue by using a multi-level approach combining large scale gene expression profiling, gene knockouts, physiology, and manipulation of neuronal activity with a behavioral model of conditioned taste aversion. This study demonstrates that specific molecules involved in neuronal activity, in basolateral amygdala neurons that send axons to other brain regions, are important for the long-term memory of taste aversion.

**Decision letter after peer review:**

[Editors’ note: the authors submitted for reconsideration following the decision after peer review. What follows is the decision letter after the first round of review.]

Thank you for submitting your work entitled "Deletion of Stk11 and Fos BLA projection neurons alters intrinsic excitability and impairs formation of aversive memory" for consideration by *eLife*. Your article has been reviewed by a Senior Editor, a Reviewing Editor, and three reviewers. The following individuals involved in review of your submission have agreed to reveal their identity: A.J. Robison (Reviewer #1); Shauna L Parkes (Reviewer #3).

Our decision has been reached after consultation between the reviewers. Based on these discussions and the individual reviews below, we regret to inform you that your work will not be considered further for publication in *eLife*.

There was agreement that the results would be of interest for the readers of *eLife*, but the major concerns noted by the reviewers require additional work that is beyond the scope of the time scale for revision for *eLife*. Given the enthusiasm for the study, if you are able to address each of the reviewers' concerns, we would be willing to reconsider your manuscript as a new submission. The major concerns regarded the lack of a cohesive story and a clear model, and the statistical analyses. These are detailed in the reviews below. In their discussion, the reviewers noted suggestions for the revision that could help to provide a clear model. It was suggested to examine whether the KO of one gene changes expression of the other gene to determine if c-Fos or Stk11 is the upstream molecule. A further suggestion was to perform a double knockout study to determine if this is additive for or occludes excitability. Overall, additional work would be needed to cohesively convey the message that learning requires gene expression and that this gene expression is involved in regulating BLA neuron excitability.

Reviewer #1:

The manuscript "Deletion of Stk11 and Fos BLA projection neurons alters intrinsic excitability and impairs formation of aversive memory" uses a combination of mouse behavior, molecular biology, pharmacology, genetic manipulation, and electrophysiology to uncover a novel role for the kinase Stk11 in the function of BLA projection neurons and their role in aversive learning. The experimental approach is strong and well thought out, and the findings are mostly robust and consistent. The lack of an overall model and incomplete interpretation is a major concern, and there are a few minor concerns with terminology.

Essential revisions:

1) There is a logical disconnect between the gene expression findings and the electrophysiological findings. After learning, c-Fos and Stk11 are oppositely regulated in the BLA: cFos is increased and Stk11 is decreased (Figure 2). However, knocking out either gene increases the excitability of the neurons (Figure 6 and Figure 7). Since the authors attempt to connect excitability directly to learning (Figure 7E), this is difficult to parse. If learning decreases excitability, that fits with the induction of cFos, but not with the decrease in Stk11. The authors should attempt to address this apparent discrepancy in the Discussion and propose a model that takes all of these data into account.

2) The potential conflicting expression data for cFos and Stk11 bring up another issue: are these changes in gene expression occurring in the same cells? One possible explanation for the apparent contradiction in point 1 above is that cFos is induced in a separate population of BLA cells from those in which Stk11 is decreased. Is it possible to determine whether this cell population is the same?

Reviewer #2:

The manuscript reports on the role of Fos and Stk11 in conditioned taste aversion learning (CTA). The study focuses on projection neurons in the basolateral nucleus of the amygdala (BLApn), which is known to contribute to CTA learning. Using RNA-seq the authors identify a number of genes that are upregulated or downregulated following CTA. They choose to focus on Fos, previously shown to increase in BLA following CTA, and Stk11, a kinase that contributes to a variety of cellular functions but has not been studies in the context of learning. CTA increases Fos expression in BLA projection neurons (as previously shown) and decreases Stk11 mRNA levels. However, the level of expression of Stk11 protein is increased following CTA indicating complexity in the signaling underlying this form of aversive learning. Using a loss of function approach, the authors show that loss of Fos has a minimal effect on CTA, while loss of Slk11 impairs learning. Additional experiments examine the effect of loss of function on BLApn excitability and examine palatability coding in the gustatory cortex (GC), which receives a prominent projection from BLA and is involved in CTA learning. While the results are interesting, the study suffers from lack of clarity and several inconsistencies in the approach and results.

Essential revisions:

1) The Abstract, Introduction and Results section lack consistency. The Abstract seems to indicate that the target of the study is Stk11 and its signaling and that CTA learning is a model of learning to investigate the molecule. Differently, the Introduction seems to indicate that CTA is the main subject and that the authors aim to investigate the cellular mechanisms for CTA. Finally, in the Results section the initial setup is to show that CTA requires protein synthesis and RNA-seq is used to identify potential targets. Across all sections the choice to focus on Stk11 is never justified, leaving the reader wondering why this molecule and not another one of the >30 identified by RNA-seq.

2) The authors interpret the impaired CTA in loss of Stk11 function experiments as indicative of an increase in the expression of the gene during CTA training. However, their results indicate that Stk11 mRNA levels decrease following CTA, therefore a loss of Stk11 would be expected to induce CTA, not prevent it. On the other hand, Stk11 protein levels increase following CTA, suggesting that it is protein synthesis, not gene expression that contributes to CTA.

3) The Fos loss of function experiment also shows some inconsistency across results. Fos increases following CTA, thus one may expect that preventing this increase would prevent CTA if Fos expression is necessary for learning. However, the results reported here show only a minor effect on CTA magnitude, suggesting that the increase in Fos expression plays a minor effect in this form of aversive learning.

4) The connection between CTA, gene expression and neuronal excitability is rather weak and indirect. If the increase in Fos is prevented neurons become more excitable, which is in contrast with the idea that Fos is induced by neuronal activation. As CTA is still induced in these conditions, the relationship between Fos, learning and neuronal excitability is unclear. In the case of Stk1, loss of function also increases excitability, but in this case there is no CTA.

5) The palatability data from GC in the Stk11 KO model are also hard to interpret. Previous data by the Katz lab showed that spiking activity of GC neurons is altered following CTA. As a loss of function in BLA Stk11 mimics the effect of CTA, one would expect that the Stk11-KO alone would alter GC activity in the palatability epoch. However, the data show otherwise, suggesting that Stk11 signaling may not be sufficient for altering palatability coding in GC. While from the signaling perspective this is an interesting finding, it is hard to reconcile with the other components of this study.

Reviewer #3:

Levitan and colleagues explore the role of the kinase Stk11 in aversive learning. I enjoyed reading this manuscript. The authors have made a considerable effort to carefully control their experiments, provide some specificity to the effect of Stk11 deletion as well as provide a potential mechanism (intrinsic BLApn excitability) for their findings. Briefly, the authors report that new transcription in the BLA is required for CTA learning and such learning causes changes in the expression of the kinase Stk11 in BLA projection neurons. Conditional Stk11 knockout in BLApn during (but not after) taste-LiCl pairings blocks learning and also increases the excitability of BLA projection neurons. This change in excitability did not appear to alter the ability of the BLA-GC circuit to respond to gustatory stimuli. These data are particularly exciting as Stk11 has not previously been implicated in learning. I believe that this manuscript is worthy of publication in *eLife* but have a few concerns that should be addressed.

1) Figure caption 3: Could the authors please clarify the statistics reported for panel C. The authors state N=8 per group (and two groups: GFP and CRE) so a total of 16 mice. However, the df for the genotype main effect (GFP vs CRE) is reported as (1,28). I don't understand how 28 is calculated for 16 animals and 2 between-subjects groups. In addition, in subsection “Fos and Stk11 expression in BLA projection neurons are necessary for memory formation”, it is stated that "Post hoc analysis (Bonferroni) revealed that both GFP (n=8) and Cre (N=8) group reductions following CTA (test vs. train) were significant but differences between other groups were not." What are these other groups? Were there initially other groups involved in this experiment? If so, I'm wondering if that might also explain the somewhat surprising significant interaction reported in (C). Finally, on Figure 3C, N=8 is labeled for each column giving the impression that the "train" and "test" conditions were not the same animals. I imagine that the N should only be reported on the "train" column, as in Figure 4A and 4D.

2) Subsection “Stk11 deletion does not impact basal aspects of taste behavior” – It wasn't immediately clear to me what "behavior" referred to in this experiment. It seems that "behavior" is the GC neurons responses rather than the behavior of the mice, is that correct? Indeed, if I understood correctly, in this manuscript, there was no behavioral measurement of palatability (e.g., licking microstructure, number of licks etc). However, if this is the case, to what does "the behaviorally-determined palatability ranking" refer? Does it refer to another data set that was previously reported (Levitan et al., 2019), as stated? If so, can the authors please justify the use of behavioral data from another set of mice (perhaps already published) to calculate the correlations (between firing rates and behaviorally measured palatability) reported in Figure 5.

3) Could the authors please provide a brief justification for using only the taste only control (and not the LiCl control) in the majority of experiments?

4) Introduction: "Not known to play a role in learning or the regulation of intrinsic neuronal excitability" It is unclear from this sentence if the authors mean that the role of Stk11 in learning/neuronal excitability has not previously been studied or if it has been studied and no evidence was found for an involvement in these processes.

---

## [Author Response]

[Editors’ note: the authors resubmitted a revised version of the paper for consideration. What follows is the authors’ response to the first round of review.]

The major concerns regarded the lack of a cohesive story and a clear model, and the statistical analyses. These are detailed in the reviews below. In their discussion, the reviewers noted suggestions for the revision that could help to provide a clear model. It was suggested to examine whether the KO of one gene changes expression of the other gene to determine if c-Fos or Stk11 is the upstream molecule. A further suggestion was to perform a double knockout study to determine if this is additive for or occludes excitability. Overall, additional work would be needed to cohesively convey the message that learning requires gene expression and that this gene expression is involved in regulating BLA neuron excitability.

We have taken the reviewers’ and editors’ suggestion to examine whether knockout of one gene changes expression of the other. We find that *Stk11* KO animals can still upregulate C-FOS expression to a similar degree and that STK11 expression is unaltered in the *Fos* KO animals (Figure 5—figure supplement 1). We conclude, that at least at the level of *Stk11* and *Fos* themselves, the genes are not upstream or downstream of one another and separately feed into mechanisms of neuronal excitability and learning. We have not ruled out possible interactions of downstream portions of the signaling cascade and make this clear in the manuscript.

We have not performed the double knockout study for several reasons. First, we calculated that it would require close to a year of additional breeding since it would require mice that carry 5 alleles (two of each knockout, plus the cre-dependent reporter) and such mice are impossible to achieve in one generation and are more likely to require 3 or 4 generations or more. Second, and perhaps more importantly, we are not convinced it would be informative. Since the blockade of learning in the case of the *Stk11* knockout is essentially complete, it would be very difficult to quantitatively distinguish between additive and occlusive phenotypes. The *Fos* KO produces a less complete blockade of learning, but in either an additive (independent pathways) or occlusive (same pathway) model, the observed effect of the double KO would be increased relative to the *Fos* KO and unchanged relative to the *Stk11* KO.

Finally, we have revised the statistical approach as suggested.

Specific replies to all of the issues raised by each of the reviewers are given below.

In a nutshell, the new data and model address the reviewer’s concerns and comments specifically regarding:

1) Potential interactions between *Stk11* and *Fos*. Figure 5—figure supplement 1 shows evidence for the lack of interaction between *Fos* and *Stk11* signaling pathways. subsection “*Fos* and *Stk11* expression in BLA projection neurons are necessary for memory formation”, Subsection “Intrinsic excitability of BLApn as a candidate mechanism for Fos and Stk11’s effects on CTA memory.

2) Causal relation between CTA memory formation and BLApn intrinsic excitability. Figure 8 (including Figure 8—figure supplement 1) shows the causal relation between CTA memory formation and BLApn intrinsic excitability. Subsection “*Stk11* and *Fos* deletion increase the intrinsic excitability of BLApn, while CTA decreases excitability”, Subsection “Necessity of Stk11 implicates the AMP-related kinase pathway in learning”.

3) A clear model that describes the ways *Stk11* and *Fos* signaling pathways effects memory formation through their modulation of intrinsic excitability. A new figure (Figure 9) now summarizes our model for the roles of Stk11, Fos and intrinsic excitability in the formation of CTA memory. Subsection “*Stk11* and *Fos* deletion increase the intrinsic excitability of BLApn, while CTA decreases excitability”; subsection “Intrinsic excitability of BLApn as a candidate mechanism for Fos and Stk11’s effects on CTA memory.”.

Reviewer #1:The manuscript "Deletion of Stk11 and Fos BLA projection neurons alters intrinsic excitability and impairs formation of aversive memory" uses a combination of mouse behavior, molecular biology, pharmacology, genetic manipulation, and electrophysiology to uncover a novel role for the kinase Stk11 in the function of BLA projection neurons and their role in aversive learning. The experimental approach is strong and well thought out, and the findings are mostly robustand consistent.

Thank you for highlighting positive features of the manuscript.

The lack of an overall model and incomplete interpretation is a major concern, and there are a few minor concerns with terminology.

In our revised manuscript we added new experiments and analyses and provide a model summarized diagrammatically (Figure 9) and explained in the Discussion section. The question of terminology is addressed below.

Essential revisions:1) There is a logical disconnect between the gene expression findings and the electrophysiological findings. After learning, c-Fos and Stk11 are oppositely regulated in the BLA: cFos is increased and Stk11 is decreased (Figure 2). However, knocking out either gene increases the excitability of the neurons (Figure 6 and Figure 7). Since the authors attempt to connect excitability directly to learning (Figure 7E), this is difficult to parse. If learning decreases excitability, that fits with the induction of cFos, but not with the decrease in Stk11. The authors should attempt to address this apparent discrepancy in the Discussion and propose a model that takes all of these data into account.

The reviewer is absolutely correct that levels of *Stk11* transcript decline at the time point measured, however we also found that levels of protein were increased at the same time point. We appreciate the suggestion of presenting a model of how this might occur and have tried to do so both in the discussion and schematically in Figure 9. Although *Stk11* mRNA decreases following CTA training (Figure 2B and C) protein levels increase (Figure 2—figure supplement 2). Although discrepancies between measurement of RNA and protein expression are unusual, they do occur. Such discrepancies would be expected if activation of Stk11 led to feedback inhibition of subsequent transcription or to enhanced mRNA degradation. These types of negative feedback have been described for other genes, a point we now discuss subsection “BLA projection neurons undergo transcription important for CTA learning”. We are actively pursuing the question of whether and how this feedback occurs, but this is likely to require a significant body of new experiments which we feel are better left for a follow up paper.

The finding that protein levels of STK11 increase are consistent with the fact that *Stk11* deletion before CTA training impairs CTA memory (Figure 4). The finding that *Stk11* deletion increases BLApn intrinsic excitability (Figure 6) is consistent with the reduction in intrinsic excitability evident after CTA training (Figure 7) and with the fact that increasing excitability using *hM3Dq* DREADD impairs CTA memory (Figure 8). This logic is now discussed in the text (subsection “BLA projection neurons undergo transcription important for CTA learning”) and diagrammed in Figure 9. Subsection “*Stk11* and *Fos* deletion increase the intrinsic excitability of BLApn, while CTA decreases excitability”; subsection “Intrinsic excitability of BLApn as a candidate mechanism for Fos and Stk11’s effects on CTA memory.”.

2) The potential conflicting expression data for cFos and Stk11 bring up another issue: are these changes in gene expression occurring in the same cells? One possible explanation for the apparent contradiction in point 1 above is that cFos is induced in a separate population of BLA cells from those in which Stk11 is decreased. Is it possible to determine whether this cell population is the same?

The issue brought up by the reviewer is interesting and important, however a complete cellular map of changes in the expression of *Fos* and *Stk11* transcript and protein is beyond the scope of this manuscript. *Stk11* is ubiquitously expressed both in humans and in mice (see e.g. https://www.tau.ac.il/~elieis/HKG/HK_genes.txt and in situ hybridization for this gene in the Allen Brain Atlas). Hence it is likely that cells activating *Fos* represent a subset of the cells activating *Stk11*. Future studies will aim to determine with a much finer resolution the distribution of these genes among the population of BLApn.

Reviewer #2:The manuscript reports on the role of Fos and Stk11 in conditioned taste aversion learning (CTA). The study focuses on projection neurons in the basolateral nucleus of the amygdala (BLApn), which is known to contribute to CTA learning. Using RNA-seq the authors identify a number of genes that are upregulated or downregulated following CTA. They choose to focus on Fos, previously shown to increase in BLA following CTA, and Stk11, a kinase that contributes to a variety of cellular functions but has not been studies in the context of learning. CTA increases Fos expression in BLA projection neurons (as previously shown) and decreases Stk11 mRNA levels. However, the level of expression of Stk11 protein is increased following CTA indicating complexity in the signaling underlying this form of aversive learning. Using a loss of function approach, the authors show that loss of Fos has a minimal effect on CTA, while loss of Slk11 impairs learning. Additional experiments examine the effect of loss of function on BLApn excitability and examine palatability coding in the gustatory cortex (GC), which receives a prominent projection from BLA and is involved in CTA learning. While the results are interesting, the study suffers from lack of clarity and several inconsistencies in the approach and results.Essential revisions:1) The Abstract, Introduction and Results section lack consistency. The Abstract seems to indicate that the target of the study is Stk11 and its signaling and that CTA learning is a model of learning to investigate the molecule. Differently, the Introduction seems to indicate that CTA is the main subject and that the authors aim to investigate the cellular mechanisms for CTA.

We thank the reviewer for pointing out that our argument was not as clearly presented as it should have been. We have rewritten portions of the Abstract, Introduction and Results section to attempt to clarify the essential argument made succinctly in our title. The molecular and cellular mechanisms of CTA within the BLA are our primary interest. The focus on *Fos* and *Stk11* and the way the two sets of results are balanced reflect the following important considerations:

1) *Fos* is known to be activated during learning and to be important for CTA specifically, hence the fact that *Fos* changes in expression is an important positive control but is not in and of itself of immediate general interest.

2) *Stk11* and the AMP-related kinase signaling pathway are highly studied (2755 and 18,635 pubmed entries respectively) and have broad significance outside of neuroscience but have not previously been implicated in learning. Demonstration that the master kinase in this pathway is important for CTA learning is an important new result and hence is emphasized.

3) Uniquely, among the genes identified, *Fos* and *Stk11* have existing conditional alleles that allow local, conditional knockout specifically within BLA projection neurons. This is a far more powerful approach for investigating gene function in learning than other more partial or more global approaches.

4) Surprisingly, both genes appear to have similar effects on BLApn excitability, leading us to focus on this cellular context for their molecular actions, and strengthening the importance of considering both genes in the paper. We have now added results that show that manipulating excitability of these neurons independently of either gene also blocks learning and have shown that the two genes independently converge on excitability since knocking out one does not affect expression of the other.

We feel that although it is more complex to include manipulations of two genes and cellular and behavioral analyses in the paper, the story is greatly enriched by this inclusion. We have tried to make the essential logic clearer with changes to the Abstract, Introduction, Results section, Discussion section and a new figure diagramming our model, but we welcome additional suggestions as to how we could further clarify the choice of genes to investigate, or other aspects of our argument.

Finally, in the results section the initial setup is to show that CTA requires protein synthesis and RNA-seq is used to identify potential targets. Across all sections the choice to focus on Stk11 is never justified, leaving the reader wondering why this molecule and not another one of the >30 identified by RNA-seq.2) The authors interpret the impaired CTA in loss of Stk11 function experiments as indicative of an increase in the expression of the gene during CTA training. However, their results indicate that Stk11 mRNA levels decrease following CTA, therefore a loss of Stk11 would be expected to induce CTA, not prevent it. On the other hand, Stk11 protein levels increase following CTA, suggesting that it is protein synthesis, not gene expression that contributes to CTA.

The initial portion of the results demonstrate that new transcription, and not only new protein synthesis are required. That new protein synthesis is required for CTA was previously known, but here we show for the first time that CTA also requires new transcription.

For a gene to affect learning it must both be transcribed and translated. We do not know that new transcription of *Stk11* is required for learning and the discussion makes note of this fact (subsection “BLA projection neurons undergo transcription important for CTA learning”). We were indeed surprised to find that the transcript and protein levels of *Stk11* are discordant at the time point measured. We have added this point to our discussion and included it in our model. We are actively investigating the biological basis of this unusual regulation. Nevertheless, the increased levels of protein are quite consistent with the fact that loss of the gene blocks learning.

3) The Fos loss of function experiment also shows some inconsistency across results. Fos increases following CTA, thus one may expect that preventing this increase would prevent CTA if Fos expression is necessary for learning. However, the results reported here show only a minor effect on CTA magnitude, suggesting that the increase in Fos expression plays a minor effect in this form of aversive learning.4) The connection between CTA, gene expression and neuronal excitability is rather weak and indirect. If the increase in Fos is prevented neurons become more excitable, which is in contrast with the idea that Fos is induced by neuronal activation. As CTA is still induced in these conditions, the relationship between Fos, learning and neuronal excitability is unclear. In the case of Stk1, loss of function also increases excitability, but in this case there is no CTA.

Genes differ in their degree of penetrance with respect to many phenotypes, including complex phenotypes like learning. The effect of *Fo*s knockout on learning is smaller in amplitude than that observed for *Stk11*, but the results are robust (2-fold increase in the fraction of Saccharin consumed vs. a 2.4 fold increase for *Stk11*) and reproducible (significant interaction between genotype and training). We do not see differences in penetrance as a concern but suggest that *Stk11* and *Fos* effect BLApn intrinsic excitability (and thus memory) by different signaling pathways and mechanisms. This is further reinforced by new results showing that *Fos* cKO does not effect STK11 expression and that *Stk11* cKO does not affect C-FOS expression (Figure 5—figure supplement 1). This is presented and discussed in: Subsection “*Fos* and *Stk11* expression in BLA projection neurons are necessary for memory formation”, subsection “Intrinsic excitability of BLApn as a candidate mechanism for Fos and Stk11’s effects on CTA memory”.

Additionally, we do not claim to have ruled out the possibility that one or both genes also affect behavior through additional mechanisms that do not depend on changes in excitability (subsection “Necessity of Stk11 implicates the AMP-related kinase pathway in learning”).

Finally, we have directly tested the relationship between learning and neuronal excitability using chemogenetic manipulation of excitability independently of manipulating either signaling pathway. These results (Figure 8 and Figure 8—figure supplement 1) make a stronger argument for causal involvement in learning of changes in excitability. Subsection “*Stk11* and *Fos* deletion increase the intrinsic excitability of BLApn, while CTA decreases excitability”; subsection “Necessity of Stk11 implicates the AMP-related kinase pathway in learning”.

5) The palatability data from GC in the Stk11 KO model are also hard to interpret. Previous data by the Katz lab showed that spiking activity of GC neurons is altered following CTA. As a loss of function in BLA Stk11 mimics the effect of CTA, one would expect that the Stk11-KO alone would alter GC activity in the palatability epoch. However, the data show otherwise, suggesting that Stk11 signaling may not be sufficient for altering palatability coding in GC. While from the signaling perspective this is an interesting finding, it is hard to reconcile with the other components of this study.

The reviewer is correct that if KO of *Stk11* produced the identical effects to those of learning, a change in GC activity in the palatability epoch would be expected. However, we do not suggest that KO of *Stk11* mimics learning. For example, the two have opposite effects on BLApn intrinsic excitability: learning decreases, while *Stk11* KO increases excitability. We were worried that *Stk11* KO might have other baseline effects on sensory processing or on the ability of BLApn to respond to training by transcriptionally activating C-FOS. If either the baseline coding or ability to become activated during training were impaired, this might explain the learning phenotype. Instead, our results show that cKO of *Stk11* does not affect GC coding of taste palatability at baseline (Figure 5) and that KO neurons in the BLA can still respond to training by activating C-FOS (Figure 5-figure supplement 1). Together, these results argue that *Stk11* is important for CTA memory formation, rather than for the initial ability of the system to respond to conditioned and unconditioned stimuli. This idea is also supported by the fact that *Stk11* KO does not influence CTA retrieval (Figure 4D).

Reviewer #3:Levitan and colleagues explore the role of the kinase Stk11 in aversive learning. I enjoyed reading this manuscript. The authors have made a considerable effort to carefully control their experiments, provide some specificity to the effect of Stk11 deletion as well as provide a potential mechanism (intrinsic BLApn excitability) for their findings. Briefly, the authors report that new transcription in the BLA is required for CTA learning and such learning causes changes in the expression of the kinase Stk11 in BLA projection neurons. Conditional Stk11 knockout in BLApn during (but not after) taste-LiCl pairings blocks learning and also increases the excitability of BLA projection neurons. This change in excitability did not appear to alter the ability of the BLA-GC circuit to respond to gustatory stimuli. These data are particularly exciting as Stk11 has not previously been implicated in learning. I believe that this manuscript is worthy of publication in eLife but have a few concerns that should be addressed.

Thank you for appreciating the positive aspects of our study.

1) Figure caption 3: Could the authors please clarify the statistics reported for panel C. The authors state N=8 per group (and two groups: GFP and CRE) so a total of 16 mice. However, the df for the genotype main effect (GFP vs CRE) is reported as (1,28). I don't understand how 28 is calculated for 16 animals and 2 between-subjects groups. In addition, in subsection “Fos and Stk11 expression in BLA projection neurons are necessary for memory formation”, it is stated that "Post hoc analysis (Bonferroni) revealed that both GFP (n=8) and Cre (N=8) group reductions following CTA (test vs. train) were significant but differences between other groups were not." What are these other groups? Were there initially other groups involved in this experiment? If so, I'm wondering if that might also explain the somewhat surprising significant interaction reported in (C). Finally, on Figure 3C, N=8 is labeled for each column giving the impression that the "train" and "test" conditions were not the same animals. I imagine that the N should only be reported on the "train" column, as in Figure 4A and 4D.

We revised all statistical analysis in the manuscript including Figure 3. For example, for Figure 3 we now report F(1,14). The degrees of freedom is 14 reflecting the 16 animals in two groups. The reviewer is correct that there are no other groups and that we were incorrect and unclear in referring to other conditions as if they were groups. This is corrected in the figure, legend and corresponding text. The interaction reported in C is expected since although both genotypes (control GFP and Cre knockout) have a reduction following training, the reduction is larger in the control animals than in the knockout, hence there is an interaction between training and genotype. We have clarified the labeling of N in Figure 3 as requested.

2) Subsection “Stk11 deletion does not impact basal aspects of taste behavior” – It wasn't immediately clear to me what "behavior" referred to in this experiment. It seems that "behavior" is the GC neurons responses rather than the behavior of the mice, is that correct? Indeed, if I understood correctly, in this manuscript, there was no behavioral measurement of palatability (e.g., licking microstructure, number of licks etc).

The reviewer is correct. We have corrected this misstatement by changing the sentence to:

“*Stk11* deletion in BLApn does not impact taste palatability coding in the GC” subsection “*Fos* and *Stk11* expression in BLA projection neurons are necessary for memory formation”.

However, if this is the case, to what does "the behaviorally-determined palatability ranking" refer? Does it refer to another data set that was previously reported (Levitan et al., 2019), as stated? If so, can the authors please justify the use of behavioral data from another set of mice (perhaps already published) to calculate the correlations (between firing rates and behaviorally measured palatability) reported in Figure 5.

The reviewer is correct. We used the palatability rankings of four taste stimuli (sucrose, sodium chloride, citric acid and quinine) previously determined by us in a recent study in mice (Levitan et al., 2019). This ranking is highly robust and has been repeated in other studies in mice (Fletcher et al., 2017; Lavi et al., 2018) and rats (Katz et al., 2001; Piette et al., 2002). The use of a nonparametric rank correlation ensures that the results are robust despite variations in absolute firing rates or preferences across animals. We refined this point in the methods section and in the Figure legend. subsection “*Stk11* deletion in BLApn does not impact taste palatability coding in the GC”, subsection “*In-vivo recording of GC taste responses”*.

3) Could the authors please provide a brief justification for using only the taste only control (and not the LiCl control) in the majority of experiments?

A single control condition consisting of the taste stimulus without the unconditioned stimulus was used in the screening experiments of Figure 1 and Figure 2 (and associated supplements) since our principle aim was to identify target genes that could then be tested for causal roles in learning in subsequent behavioral experiments. A prior study from the Rosenblum lab (Rappaport et al., 2015) used a microarray screen following CTA to show that LiCl treatment (compared to taste control treatment) induced more than half of the same genes induced by CTA treatment (compared to taste control treatment). We have now made this point clear in the text. subsection “Transcripts regulated in BLA projection neurons following CTA”.

4) Introduction: "Not known to play a role in learning or the regulation of intrinsic neuronal excitability" It is unclear from this sentence if the authors mean that the role of Stk11 in learning/neuronal excitability has not previously been studied or if it has been studied and no evidence was found for an involvement in these processes.

We meant the former. We have clarified this by changing the wording to: “it has not previously been studied in the context of learning and memory or in the regulation of intrinsic neuronal excitability.”